# FlowMo: Variance-Based Flow Guidance for Coherent Motion in Video Generation

**Ariel Shaulov**[*]    **Itay Hazan**[*]    **Lior Wolf**    **Hila Chefer**
School of Computer Science
Tel Aviv University, Israel

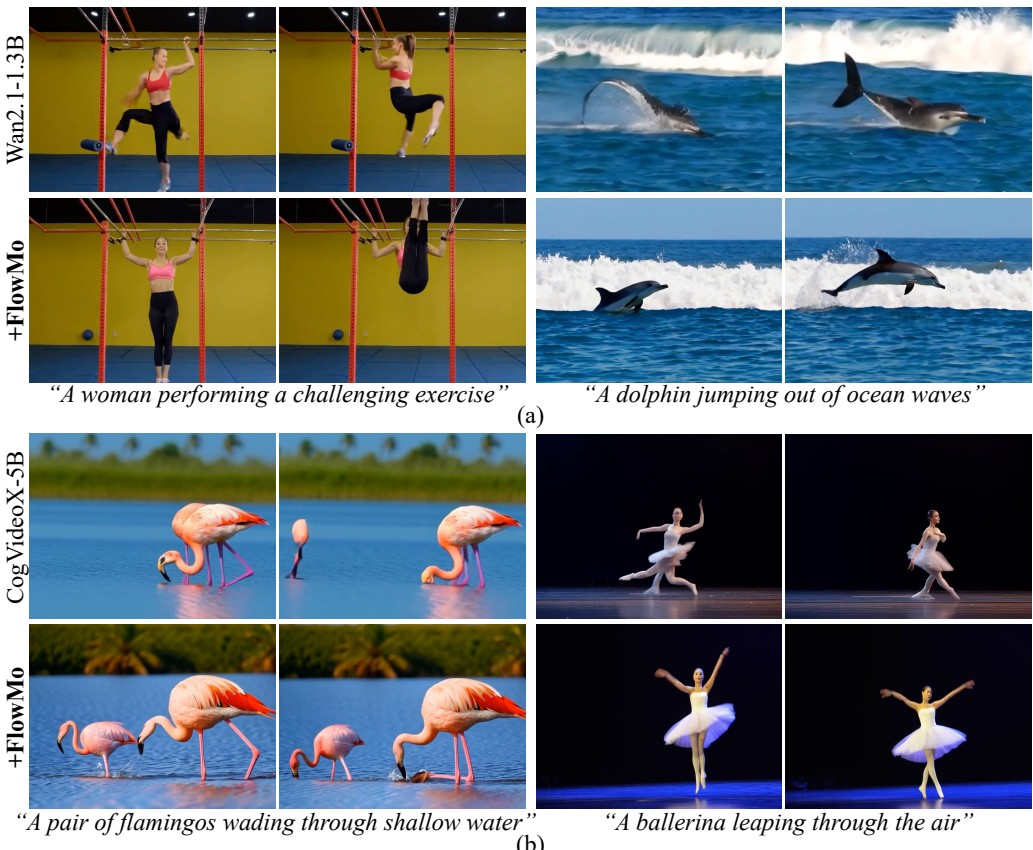

*"A woman performing a challenging exercise"*          *"A dolphin jumping out of ocean waves"*

(a)

*"A pair of flamingos wading through shallow water"*          *"A ballerina leaping through the air"*

(b)

Figure 1: **Text-to-video results before and after applying FlowMo on (a) Wan2.1 [1] and CogVideoX-5B [2]**. We present *FlowMo*, an inference-time guidance method to enhance temporal coherence in text-to-video models. Our method mitigates severe temporal artifacts, such as additional limbs (woman, 1st row, 2nd row), objects that appear or disappear (flamingo, 2nd row), and object distortions (woman, dolphin, 1st row), without requiring additional training or conditioning signals.

## Abstract

Text-to-video diffusion models are notoriously limited in their ability to model temporal aspects such as motion, physics, and dynamic interactions. Existing approaches address this limitation by retraining the model or introducing external conditioning signals to enforce temporal consistency. In this work, we explore

---

[*]Equal contribution.
Project page: `https://arielshaulov.github.io/FlowMo/`
Correspondence to: Ariel Shaulov: `arielshaulov@mail.tau.ac.il`, Itay Hazan: `itay.hzn@gmail.com`.

39th Conference on Neural Information Processing Systems (NeurIPS 2025).

whether a meaningful temporal representation can be extracted directly from the predictions of a pre-trained model without any additional training or auxiliary inputs. We introduce **FlowMo**, a novel training-free guidance method that enhances motion coherence using only the model's own predictions in each diffusion step. FlowMo first derives an appearance-debiased temporal representation by measuring the distance between latents corresponding to consecutive frames. This highlights the implicit temporal structure predicted by the model. It then estimates motion coherence by measuring the patch-wise variance across the temporal dimension and guides the model to reduce this variance dynamically during sampling. Extensive experiments across multiple text-to-video models demonstrate that FlowMo significantly improves motion coherence without sacrificing visual quality or prompt alignment, offering an effective plug-and-play solution for enhancing the temporal fidelity of pre-trained video diffusion models.

# 1    Introduction

Despite recent progress, text-to-video diffusion models remain far from faithfully capturing the temporal dynamics of the real world. Generated videos frequently exhibit temporal artifacts such as objects appearing and disappearing, duplicated or missing limbs, and abrupt motion discontinuities [3, 4, 5]. These issues highlight the limited capability of text-to-video models to reason about motion, physics, and dynamic interactions over time. To mitigate these shortcomings, prior works have proposed fine-tuning models with explicit motion-related objectives [3], conditioning the generation on external motion signals such as optical flow or pixel trajectories [6, 7, 8, 9], or designing complex model architectures tailored to capture temporal dependencies [10, 11, 12].

However, these approaches require either retraining the model [3, 11, 12] or introducing rigid external constraints that dictate motion [6, 7], limiting flexibility and generality. In this work, we propose an alternative strategy dubbed **FlowMo**, a training-free guidance method that improves temporal consistency using the model's own internal representations during sampling. FlowMo extracts a latent temporal signal directly from the pre-trained model during inference and leverages its statistics to derive a guidance signal, without any architectural modifications, training, or external supervision.

Our method is grounded in the following key observation: the temporal evolution of individual spatial patches tends to be smooth when the motion is coherent. Namely, the shifts in the representation of each patch over time are expected to be relatively small, leading to low patch-wise variance across frames. In contrast, incoherent motion intuitively manifests as abrupt changes in appearance or structure, producing high temporal variance in the patches that display temporal artifacts.

Notably, measuring temporal relations in a way that is disentangled from appearance information is challenging. As observed by previous works [3], the predictions of text-to-video models are biased toward appearance-based features. To obtain a meaningful appearance-debiased representation, we use the model's latent predictions to compute pairwise distances between frames. This enables us to measure the shifts in patch representations using patch-wise variance over time while neutralizing their shared appearance content. This is motivated by prior works demonstrating that the latent spaces of generative models capture semantically meaningful transformations, where simple vector operations correspond to interpretable changes [13, 14, 15].

We explore the above intuition extensively in Sec. 3.2. First, we collect a set of generated videos that exhibit significant motion. We categorize these videos into coherent and incoherent sets, and compute the patch-based variance over time given the appearance-debiased representations discussed above. Our experiments yield two complementary observations. First, we find a clear correlation between high patch-based variance and motion incoherence, indicating that measuring the shift in patch representations over time can serve as a reliable metric to estimate coherence. Second, we observe both qualitatively and quantitatively that while coarse appearance-based features such as scene layout and spatial structure are established very early in the generation process, temporal information emerges only at later, intermediate denoising steps.

Motivated by these findings, we present FlowMo, a method that dynamically guides text-to-video diffusion models toward temporally coherent generations. At selected timesteps in the denoising process, we compute the maximal patch-wise variance over time, given the appearance-debiased latent prediction. We then optimize the model's prediction to reduce this temporal variance, thereby

encouraging smoother, more coherent motion. This guidance is applied iteratively across timesteps, allowing FlowMo to influence both coarse and fine motion dynamics in the generation process.

We demonstrate our method's effectiveness on two of the most popular open-source models, Wan2.1-1.3B [1] and CogVideoX-5B [2]. Across a wide range of metrics, we evaluate the impact of our method on motion quality, overall video quality, and prompt alignment, using both the automatic evaluation metrics proposed by VBench [16] and human-based assessments. In all cases, we find that FlowMo consistently and significantly improves the temporal coherence of the generated videos, while preserving the aesthetic quality, text alignment, and motion magnitude (see Fig. 1).

Our results show that it is possible to extract meaningful temporal signals from the learned latent representations of text-to-video models. Such signals not only encapsulate the temporal structure of the generated videos but also serve as actionable guidance cues.

## 2   Related Work

**Text-to-video generation**    Diffusion models have revolutionized visual content creation, starting with image generation [17, 18, 19] and rapidly expanding to diverse applications such as text-to-image synthesis [20, 21] and image editing [22, 23, 24, 25, 26, 27, 28, 29, 30, 31]. This success has spurred their adoption for video generation, from cascaded diffusion models [32, 33, 34, 35, 36, 37, 38, 39, 40, 41], to most recently Diffusion Transformers (DiT) [42, 43] based on Flow Matching (FM) [44, 45, 46], which constitute the current state-of-the-art, and form the basis of our work.

**Inference-time guidance**    has emerged as a powerful technique to steer and refine the outputs of generative models across various tasks without training [47, 48, 49, 28, 50]. Such methods typically optimize the model predictions based on an auxiliary loss. Inference-time guidance for video generation has only recently emerged as a promising research vector [51, 52]. While our work also explores inference-time optimization for video generation, existing objectives and guiding signals inherently differ from ours. Li et al. [51] focus on steering video models using external motion priors, which requires access to additional motion-specific inputs, while Wei et al. [52] propose to minimize a global 3D variance loss. In contrast, our method leverages the internal latent-space dynamics to perform guidance without any auxiliary networks, perceptual objectives, or task-specific priors, making it a lightweight and fully self-supervised plug-and-play module.

**Improving temporal coherence in video generation**    Temporal coherence remains a core challenge in video synthesis [3, 4, 5, 12], and existing solutions generally fall into three categories. First, *training with temporal objectives* [3, 11, 53, 54], which improves consistency but demands significant compute and access to training data. Second, *guiding the generation with external motion signals* such as optical flow or trajectories [6, 7, 8, 9], which enforce coherence but require external inputs and are restricted to the conditioning motion. Third, *architectures designed for temporal modeling* [10, 11, 55, 56, 57, 58], which are often complex and not easily applied to pre-trained models. In contrast, FlowMo improves temporal coherence directly at inference time by leveraging the model's internal representations, without additional data, inputs, or retraining.

Closest to our work, FreeInit [59] and VideoGuide [60] propose methods to reduce spatio-temporal incoherence in video generation. However, both were designed for earlier UNet-based models trained with DDPM or DDIM samplers [61, 45], which suffered from severe signal-to-noise ratio (SNR) mismatches between training and inference [59]. In contrast, modern Transformer-based FM architectures are substantially more robust, rendering these techniques less effective. For completeness, we include a comparison to FreeInit (which can be reasonably adapted to DiTs) in App. A. In our experiments, we find that applying FreeInit to DiTs results in a drop in key metrics such as the overall video quality, as well as a significant drop in the amount of generated motion.

## 3   Method

### 3.1   Preliminaries: Flow Matching in a VAE Latent Space

Following common practice in state-of-the-art image and video generation models [62, 63, 1], we consider models that leverage FM [44] to define the objective function and operate in the learned

latent space of a Variational Autoencoder (VAE) for efficiency. The VAE consists of an encoder-decoder pair $(\mathcal{E}, \mathcal{D})$, where $\mathcal{E}$ maps input data $x \sim \mathcal{X}$ from the pixel space to a lower-dimensional latent representation $z = \mathcal{E}(x) \in \mathcal{Z}$, and $\mathcal{D}$ yields a reconstruction $x \approx \mathcal{D}(z)$. Given a pre-trained VAE, FM learns a transformation from a standard Gaussian distribution in latent space $z_0 \sim \mathcal{N}(0, I)$, to a target distribution $z_1$ observed from applying $\mathcal{E}$ on the data.

At each training step, FM draws a timestep $t \in [0, 1]$, and obtains a noised intermediate latent by interpolating between $z_0$ and $z_1$, namely $z_t = (1 - t) \cdot z_1 + t \cdot z_0$. The model $u_\theta$ is then optimized to predict the velocity $v_t = \frac{dz_t}{dt} = z_0 - z_1$, namely:

$$\mathcal{L}_{\text{FM}} = \mathbb{E}_{x_1, t \sim \mathcal{U}(0,1), z_0 \sim \mathcal{N}(0,I)} \left[ \|u_\theta(z_t, t) - (z_0 - z_1)\|^2 \right]. \tag{1}$$

Once trained, samples can be generated from an initial noisy latent $z_0 \sim \mathcal{N}(0, I)$ by applying a sequence of denoising steps over a discrete schedule. At time $t_i$, $z_{t_i}$ is denoised to produce $z_{t_{i+1}}$ by applying $z_{t_{i+1}} = (1 - \sigma_{t_i}) \cdot z_{t_i} - \sigma_{t_i} \cdot u_\theta(z_{t_i}, t_i)$, where $\sigma_{t_i}$ is an interpolation coefficient determined by the scheduler.

## 3.2 Motivation

In the following, we conduct qualitative and quantitative experiments to motivate the construction of FlowMo. The experiments in this section are conducted on Wan2.1-1.3B [1] for efficiency.

We begin by describing the latent representation on which FlowMo operates. As mentioned in Sec. 3.1, at each denoinsing step, the model prediction $u_{\theta,t} := u_\theta(z_t, t)$ is an estimate of the velocity $v_t$, which represents the direction from the noise distribution to the latent space distribution. To extract a temporal representation from the prediction, we propose a *debiasing operator* $\Delta$, which computes the $\ell_1$-distance between consecutive latent frames to eliminate their common appearance information. Formally, $\Delta \colon \mathbb{R}^{F \times W \times H \times C} \to \mathbb{R}^{(F-1) \times W \times H \times C}$ is defined as:

$$\forall f \in [F-1], \forall w \in [W], \forall h \in [H], \forall c \in [C] \quad (\Delta u_{\theta,t})_{f,w,h,c} = \|(u_{\theta,t})_{f+1,w,h,c} - (u_{\theta,t})_{f,w,h,c}\|_1. \tag{2}$$

Next, we describe the motivational experiments conducted to examine the statistical characteristics of our proposed latent space.

**Quantitative motivation.** Our central hypothesis is that temporally coherent motion corresponds to a form of local stability in $u_{\theta,t}$. Specifically, in videos with smooth and consistent motion, object trajectories evolve gradually, yielding lower temporal variance in $u_{\theta,t}$. Incoherent motion, in contrast, introduces abrupt changes, manifesting as larger fluctuations and higher patch-wise variance in the latent predictions. Formally, given $u_{\theta,t}$, we define its *temporal patch-wise variance tensor* $\sigma^2 \in \mathbb{R}^{W \times H \times C}$ as the variance across frames per patch and channel, i.e. $\forall w \in [W], \forall h \in [H], \forall c \in [C]$,

$$\sigma^2_{w,h,c} = \mathbb{V}_{f \sim [F-1]} \left[ (\Delta u_{\theta,t})_{f,w,h,c} \right], \tag{3}$$

where $\mathbb{V}(X) = \mathbb{E}[(X - \mathbb{E}[X])^2]$.

To empirically validate our hypothesis, we conducted a user study wherein several hundred generated videos were rated on a 1-5 scale for both coherence and perceived amount of motion (higher is more mo-

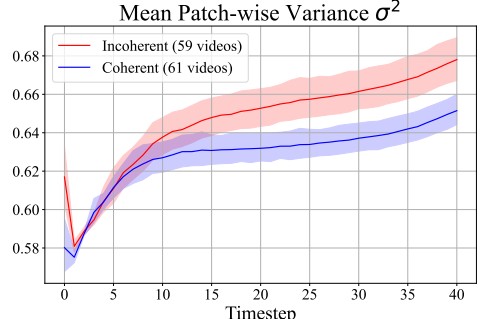

Figure 2: **Quantitative motivation**. We measure the mean temporal variance of spatial patches for coherent and incoherent videos. Incoherent videos portray higher variance. The separation is visible from step 5 onward. 95%-confidence interval was computed using the `seaborn` python package.

tion/better coherence). To isolate the effects of motion magnitude on video coherence, we focused on videos with a substantial amount of motion (rated $\geq 3$), and compared those labeled as completely incoherent (1), or completely coherent (5). As illustrated in Fig. 2, a clear negative correlation emerges: low-coherence videos consistently exhibit higher variance. This supports our intuition that temporal patch-wise variance is a meaningful measure of perceived coherence.

Notably, the separation in variance becomes prominent from approximately the fifth generation timestep onward. Next, we wish to conduct a qualitative experiment to motivate this phenomenon.

**Qualitative motivation.** To qualitatively explore the process of motion generation in text-to-video models, we visualize the evolution of the model's latent space prediction across the generation steps.

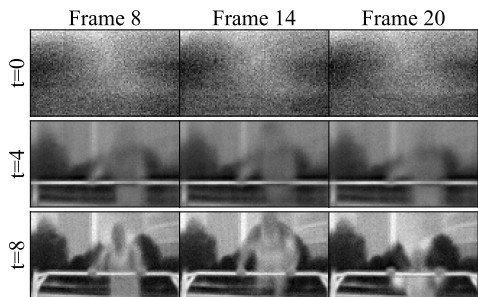

First, observe that by Sec. 3.1, the model prediction $u_{\theta,t}$ estimates the velocity $v_t = z_0 - z_1$. We can thus obtain an estimation of the fully denoised latent, $\bar{z}_1$, at any intermediary step $t$ as follows:

$$\bar{z}_1 = z_t - \sigma_t \cdot u_{\theta,t}, \tag{4}$$

where $\sigma_t$ is the signal-to-noise ratio at step $t$, which is defined by the noise scheduler.

Note that $\bar{z}_1$ is a *latent-space representation* where channels do not represent RGB information. We thus arbitrarily select the first channel, to obtain $\bar{z}_{1,c0} \in \mathbb{R}^{F \times H \times W}$, and visualize the grayscale video via:

$$V_{\bar{z}_{1,c0}} = 255 \cdot \frac{\bar{z}_{1,c0} - \min_{F,H,W}(\bar{z}_{1,c0})}{\max_{F,H,W}(\bar{z}_{1,c0}) - \min_{F,H,W}(\bar{z}_{1,c0})}. \tag{5}$$

Figure 3: **Qualitative motivation.** We visualize the model prediction per timestep across the generation. Coarse spatial information is determined in the first steps (0-4), whereas motion is determined around steps 5-8, and refined in later steps.

The resulting visualizations of $V_{\bar{z}_{1,c0}}$ in representative timesteps are presented in Fig. 3. As can be observed, coarse spatial information begins to emerge from the first generation step, where the training bar and some of the outline of the person are visible. By step 4, most of the structure of the scene is already determined. Conversely, the motion appears to be added to the scene between steps 4 and 8, as evidenced by the strong similarity between all frames in times 0,4, whereas step 8 shows significant variance between these same frames (e.g., the person bends down between frames 14 and 20). Visualizations on additional timesteps and latent channels are provided in Appendix B. Note that the above supports the quantitative experiment presented in Fig. 2. Since motion emerges only around the later initial steps of the denoising process (4-8), we would expect our variance-based metric to be meaningful only in the steps that depict measurable differences over time.

In App. C, we show quantitative evidence that, in addition to enhancing coherence, FlowMo reduces the variance in generation steps that correspond to the above intuition.

### 3.3 FlowMo

Motivated by the previous section, Algorithm 1 outlines the FlowMo guidance mechanism, applied within a single FM denoising step. We perform the FlowMo guidance at specific timesteps $\{\tau_1, \ldots, \tau_\ell\}$, corresponding to the early-to-mid stages of generation, following the motivation presented in Fig. 3.

Each denoising step $t_i$ begins by obtaining the model prediction $u_{\theta,t_i}$, given an input text prompt $\mathcal{P}$. To encourage alignment between the prediction and the textual prompt, *Classifier-free guidance* (CFG) [64] is first employed with a scale of $\rho$ (Line 3).

If we are not in a refinement step, we jump to Line 14, in which we perform a standard FM step to obtain the next latent $z_{t_{i+1}}$ as a linear combination of the current latent and the predicted velocity:

$$z_{t_{i+1}} = (1 - \sigma_{t_i}) \cdot z_{t_i} - \sigma_{t_i} \cdot u_{\theta,t_i}, \tag{6}$$

where $\sigma_{t_i}$ is a time-dependent coefficient representing the signal-to-noise ratio.

If, however, $t_i \in \{\tau_i\}_1^\ell$, a FlowMo refinement step is performed (Line 5 to Line 12). We first compute the appearance-debiased representation, $\Delta u_{\theta,t_i}$, as defined in Equation (2) (Line 5).

Subsequently, drawing on the motivation presented in Fig. 2, we calculate the temporal variance $\sigma^2_{w,h,c}$ for each spatial patch $(w, h, c)$ as defined in Equation (3) (Line 6). These patch-wise variances

are then averaged across the channel dimension to produce a single spatial map $s_{w,h}$ indicating a motion coherence score per patch (Line 7):

$$\forall w \in [W], \forall h \in [H] \quad s_{w,h} = \mathbb{E}_{c \sim [C]} \left[ \sigma^2_{w,h,c} \right] = \frac{1}{C} \sum_{c=1}^{C} \sigma^2_{w,h,c}. \tag{7}$$

The final FlowMo loss $\mathcal{L}$ is then determined by the maximal value in this map, thereby targeting the most dynamically-incoherent patch (Line 8):

$$\mathcal{L} = \max_{w \sim [W], h \sim [H]} s_{w,h}. \tag{8}$$

Intuitively, this formulation encourages the model to produce a prediction $u_{\theta, t_i}$ wherein the latent space distances of each spatial patch over time are smoother, resulting in more coherent and gradual transitions in the generated video.

Inspired by existing guidance mechanisms that optimize spatial information [47], we propose to use the loss in Eq. 7 to optimize the input latent to the diffusion step, $z_{t_i}$ (Line 9). Intuitively, this allows our optimization to modify low-level features in the generated video, including the coarse motion. Thus, the optimization is performed as a gradient descent step:

$$z_{t_i} = z_{t_i} - \eta \cdot \nabla_{z_{t_i}} \mathcal{L}, \tag{9}$$

where $\eta$ is the learning rate. Following this refinement, we repeat the denoising step $t_i$ with the optimized latent $z_{t_i}$ (Lines 10 to 12).

---

**Algorithm 1** A Single FlowMo Denoising Step

---

**Input:** A text prompt $\mathcal{P}$, a timestep $t_i$, a set of iterations for refinement $\{\tau_1, \ldots, \tau_\ell\}$, and a trained Flow Matching model $FM$.
**Output:** A noised latent $z_{t_{i+1}}$ for the next timestep $t_{i+1}$

1:   $u_{\theta, t_i | \mathcal{P}} \leftarrow FM(z_{t_i}, t_i, \mathcal{P})$
2:   $u_{\theta, t_i | \emptyset} \leftarrow FM(z_{t_i}, t_i, \emptyset)$
3:   $u_{\theta, t_i} \leftarrow u_{\theta, t_i | \emptyset} + \rho \cdot (u_{\theta, t_i | P} - u_{\theta, t_i | \emptyset})$
4: **if** $t_i \in \{\tau_1, \ldots, \tau_\ell\}$ **then**
5:     Compute $(\Delta u_{\theta, t_i})$ as in Equation (2)
6:     Compute $\sigma^2$ as in Equation (3)
7:     $s_{w,h} \leftarrow \mathbb{E}_{c \sim [C]} \left[ \sigma^2_{w,h,c} \right] \quad \forall w \forall h$
8:     $\mathcal{L} \leftarrow \max_{w \sim [W], h \sim [H]} s_{w,h}$
9:     $z_{t_i} \leftarrow z_{t_i} - \eta \cdot \nabla_{z_{t_i}} \mathcal{L}$
10:    $u_{\theta, t_i | \mathcal{P}} \leftarrow FM(z_{t_i}, t_i, \mathcal{P})$
11:    $u_{\theta, t_i | \emptyset} \leftarrow FM(z_{t_i}, t_i, \emptyset)$
12:    $u_{\theta, t_i} \leftarrow u_{\theta, t_i | \mathcal{P}} + \rho \cdot \left( u_{\theta, t_i | \mathcal{P}} - u_{\theta, t_i | \emptyset} \right)$
13: **end if**
14: $z_{t_{i+1}} \leftarrow (1 - \sigma_{t_i}) \cdot z_{t_i} - \sigma_{t_i} \cdot u_{\theta, t_i}$
15: **Return** $z_{t_{i+1}}$

---

## 4 Experiments

We conduct qualitative and quantitative experiments to demonstrate FlowMo's effectiveness. Our experiments evaluate the improvement in temporal coherence enabled by our method, as well as its ability to maintain or even enhance other aspects of the generation, such as appearance quality and text alignment. We provide our code and a website with video results in the supplemental materials.

**Implementation details** We employ two of the most popular publicly available text-to-video models: Wan2.1-1.3B [1] and CogVideoX-5B [2], using their officially provided weights and default configurations. Motivated by the insights from Sec. 3.2, we apply FlowMo in the first 12 timesteps of the generation, since these are responsible for coarse motion and structure. All our experiments employ a learning rate of $\eta = 0.005$, using the Adam optimizer, on two NVIDIA H100 GPUs, with 80GB memory each. Wan2.1 is evaluated at a resolution of $480 \times 832$, and CogVideoX at $480 \times 720$, both generating 81 frames at 16 frames per second, resulting in 5-second videos.

### 4.1 Qualitative Results

Figures 1, 4 contain representative results demonstrating the impact of FlowMo on pre-trained text-to-video models. As can be observed, our method mitigates severe temporal artifacts that are common to text-to-video models. For example, the generations tend to display extra limbs (women in Fig. 1(a) and Fig. 4, 2nd, 3rd row), distortions of objects over time (dolphin in Fig. 1(a) and deer in Fig. 4, 4th row), and objects that suddenly appear or disappear (flamingo in Fig. 1(b) and rope, violin in Fig. 4, 2nd, 3rd row). These results demonstrate that temporal artifacts correspond to abrupt changes in the latent representations of video patches. This, in turn, drives our optimization process to encourage smoother representations of the affected patches, resulting in improved temporal coherence.

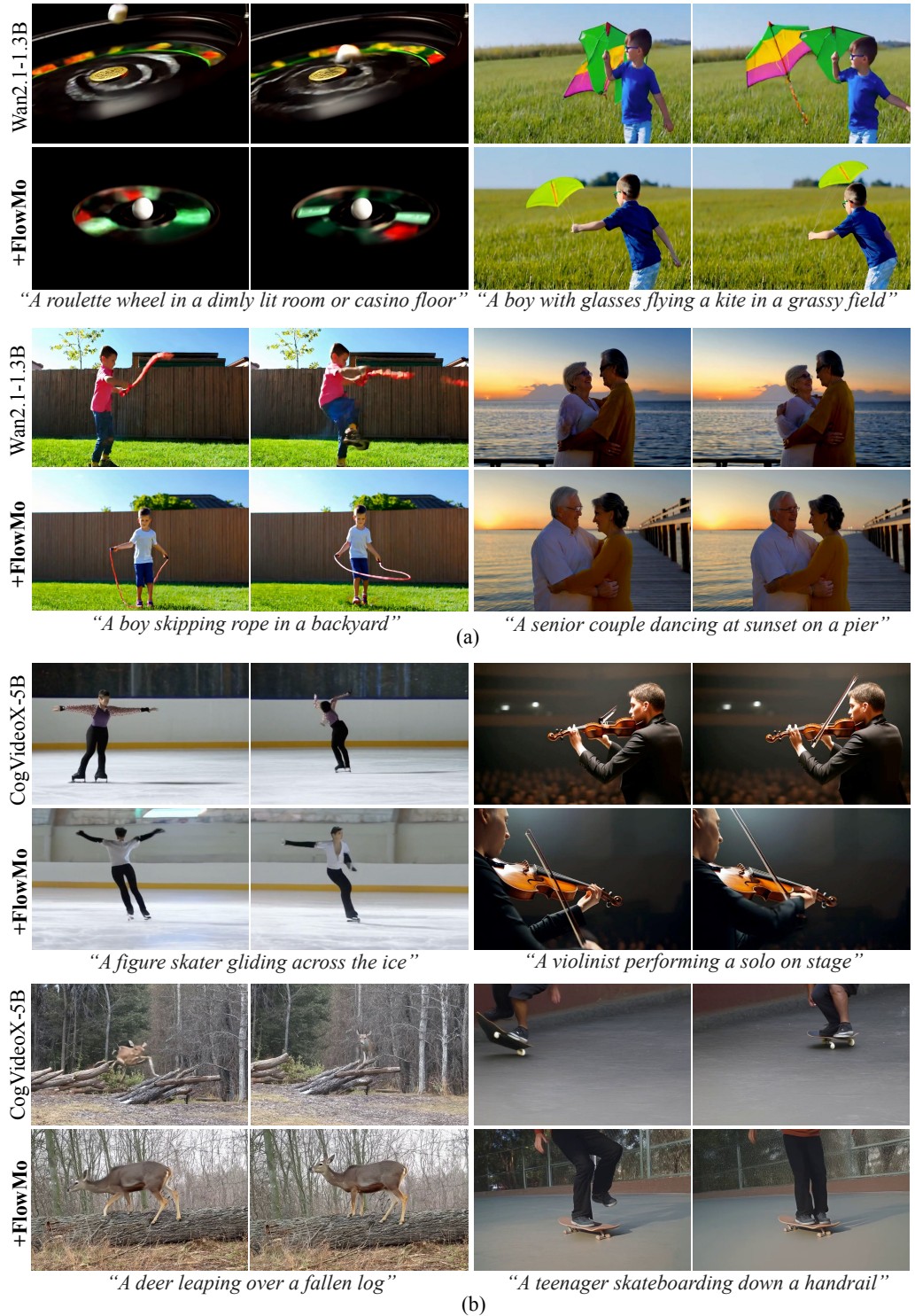

Figure 4: **Qualitative results**. Text-to-video results before and after applying FlowMo on (a) Wan2.1 [1] and (b) CogVideoX [2]. FlowMo mitigates severe temporal artifacts, e.g., extra limbs (women, 2nd, 3rd row), objects that appear or disappear (2nd, 3rd row), and distortions (4th row).

## 4.2 Quantitative Results

We employ both the VBench benchmark [16] and human-based evaluations, which serve as the standard evaluation protocols for measuring the quality of text-to-video generation [2, 65, 66, 67, 68].

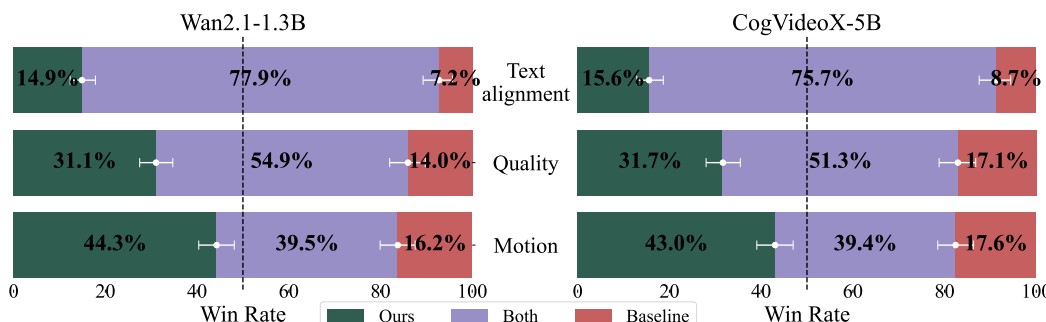

Figure 5: **User study** conducted on Wan2.1-1.3B [1] (left) and CogVideoX-5B [2] (right) using VideoJAM-bench [3], designed specifically to evaluate motion coherence. Our method significantly improves temporal coherence in all models, while maintaining or improving the visual quality and the text alignment of the resulting videos. 95%-confidence intervals were calculated using Dirichlet sampling, assuming a multinomial distribution with Laplace smoothing applied to the counts.

Table 1: **VBench evaluation results.** A comparison of the overall video quality before and after applying FlowMo on Wan2.1-1.3B [1] and CogVideoX-5B [2] using VBench [16]. We enclose both the motion-specific and the aggregated scores. FlowMo consistently improves the **Final Score** representing the overall video quality by at least 5%.

| | **Motion Metrics** | | **Aggregated Scores** | | |
| **Models** | Motion Smoothness | Dynamic Degree | Semantic Score | Quality Score | **Final Score** |
|---|---|---|---|---|---|
| Wan2.1-1.3B | 96.43% | **83.21%** | 84.70% | 65.58% | 75.14% |
| **+ FlowMo** | **98.56%** | 81.96% | **89.11%** | **73.58%** | **81.34%** (+6.20%) |
| CogVideoX-5B | 95.01% | **65.29%** | **70.03%** | 60.83% | 65.43% |
| **+ FlowMo** | **97.29%** | 63.92% | 69.26% | **72.11%** | **70.69%** (+5.26%) |

**User study.** We conduct a human preference study using the VideoJAM benchmark [3], which was specifically designed to test motion coherence. For each prompt, we generate a pair of videos (with and without FlowMo) with a fixed seed in the setting described above, and randomly shuffle the order of the results. Each prompt was evaluated by five different participants, resulting in 640 unique responses per baseline. Annotators were asked to compare the videos based on their alignment to the text prompt, the aesthetic quality of the videos, and the motion quality (see App. D).

The results, presented in Fig. 5, demonstrate a consistent human preference for FlowMo-guided videos across all criteria. Specifically for **Motion Coherence**, FlowMo was favored in 44.3% of comparisons for Wan2.1 (vs. 16.2% for baseline) and 43.0% for CogVideoX (vs. 17.6% for baseline). A similar trend was observed for **Aesthetic Quality**, where FlowMo was preferred in 31.1% of Wan2.1 pairs (vs. 14.0% for baseline) and 31.7% of CogVideoX pairs (vs. 17.1% for baseline). Interestingly, FlowMo also showed improved **Text-Video Alignment**, with preference rates of 14.9% for Wan2.1 (vs. 7.2% for baseline) and 15.6% for CogVideoX (vs. 8.7% for baseline).

These findings highlight that FlowMo not only enhances temporal coherence but also contributes positively to the overall perceived video quality and faithfulness to the input prompt.

**Automatic metrics.** The results of the automatic metrics on the VBench benchmark [16] are summarized in Tab. 1. We enclose both the motion-based metrics, and the aggregated metrics, which constitute an average of all the benchmark dimensions, and measure the overall quality of the generations. A full breakdown of all metrics is provided in App. E.

Notably, FlowMo significantly improves the **Final Score** by 6.2%, 5.26% for Wan, CogVideoX, respectively. This metric represents the overall quality score, considering all the evaluation dimensions. This improvement is supported by gains in the Quality Score (Wan2.1: +8.0%; CogVideoX: +11.28%) and Semantic Score for Wan2.1 (+4.41%), with a negligible decrease of 0.77% for CogVideoX.

Considering the motion metrics, FlowMo boosts **Motion Smoothness** (Wan2.1: +2.13%; CogVideoX: +2.28%), which is a key metric that evaluates the motion coherence. Finally, note that some decrease

to the dynamic degree is expected. This is since temporal artifacts such as objects appearing and disappearing increases the amount on motion in the video.

In summary, both human evaluations and automated VBench metrics consistently demonstrate FlowMo's effectiveness in improving motion coherence and overall video quality.

## 4.3 Comparison with FreeInit

We compare FlowMo to FreeInit [59], the most conceptually related method. The full comparison between FlowMo and FreeInit is provided in App. A.

A qualitative comparison between FlowMo and the adapted FreeInit is presented in Fig. 6. These examples highlight FlowMo's superiority in generating visually coherent motion. For instance, in the top-left example, FlowMo successfully generates a plausible marching motion, whereas FreeInit produces disappearing feet and less convincing movement. Similarly, in the top-right example, FlowMo depicts coherent walking, while the rendition produced by FreeInit suffers from partially disappearing feet and a less natural gait. The bottom-left example shows FlowMo maintaining the integrity of the man and rope, while in the video refined with FreeInit, the rope distorts and disappears and the man has a less stable form. Finally, in the bottom-right example, FlowMo maintains a consistent orientation, whereas the front and back sides of the person flip spontaneously in the version generated with FreeInit. Overall, these visual examples show that FlowMo demonstrates a significant advantage in producing more coherent and artifact-free motion compared to FreeInit.

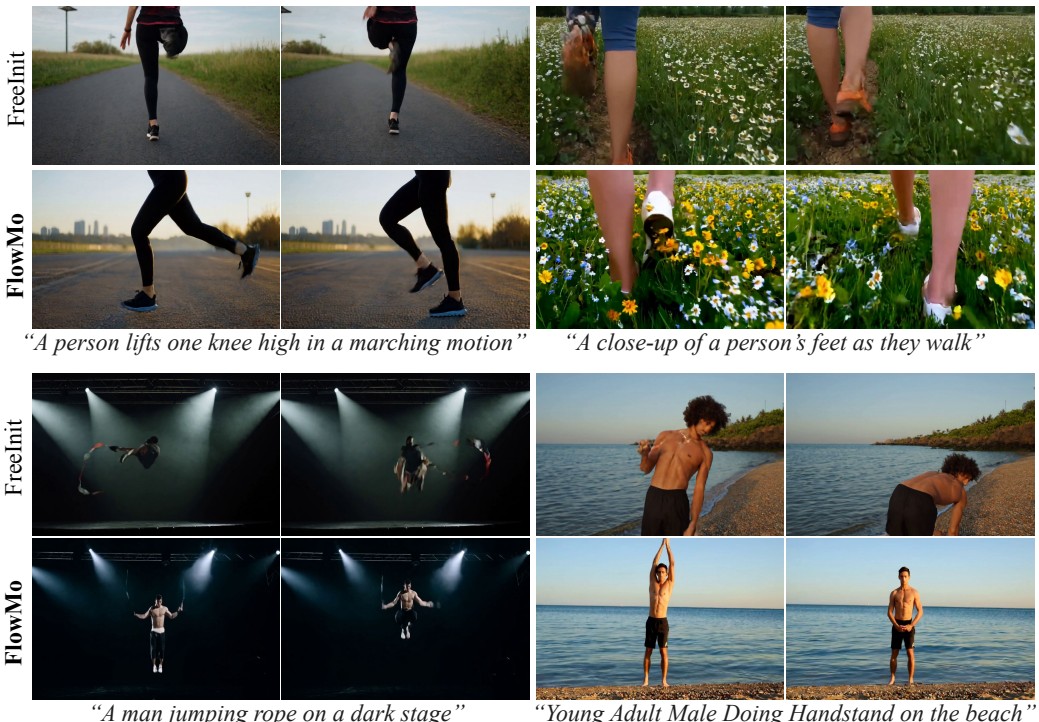

*"A person lifts one knee high in a marching motion"*    *"A close-up of a person's feet as they walk"*

*"A man jumping rope on a dark stage"*    *"Young Adult Male Doing Handstand on the beach"*

Figure 6: **Qualitative results.** Text-to-video results of FreeInit [59] (1st, 3rd row) and FlowMo (2nd, 4th row) when applied on Wan2.1-1.3B [1]. FlowMo better mitigates severe temporal artifacts, e.g. distortions and object that appear and disappear.

We further compare FlowMo to the adapted FreeInit method through a human preference study and quantitative benchmarks. Human evaluators consistently favored FlowMo-guided videos over FreeInit and the baseline across Motion Coherence (38.7% vs. 23.4%), Aesthetic Quality (28.1% vs. 14.8%), and Text-Video alignment (16.5% vs. 5.2%). Quantitative results on VBench corroborate these findings: FlowMo improves Motion Smoothness by +2.13% and Overall Quality and Semantic Scores by +7.36% and +3.20%, respectively, achieving a Final Score of 81.34% compared to FreeInit's 75.06%. Full per-dimension comparisons, and the full implementation details are provided in App. A.

## 4.4 Ablation Study

We ablate the primary design choices of FlowMo, namely using the patch with the maximal variance in the loss (Eq. 8), the appearance debiasing operator (Eq. 2), and the choice of steps to apply the optimization (1-12).

Results of the ablation study on Wan2.1 are reported in Fig. 7. For each prompt, we enclose the result with FlowMo (1st row), Wan2.1 (2nd row) and the ablations (3rd-5th row). Replacing the maximum with the mean (3rd row) significantly weakens the effect, likely because most patches are static, leading to a smaller loss and diminished gradients. Removing the debiasing operator (4th row) yields a similar effect. This can be attributed to the fact that, as observed by previous works [3], predictions by text-to-video models tend to be appearance-based, reducing the influence of motion on the loss. Finally, applying FlowMo across all steps (5th row) in-

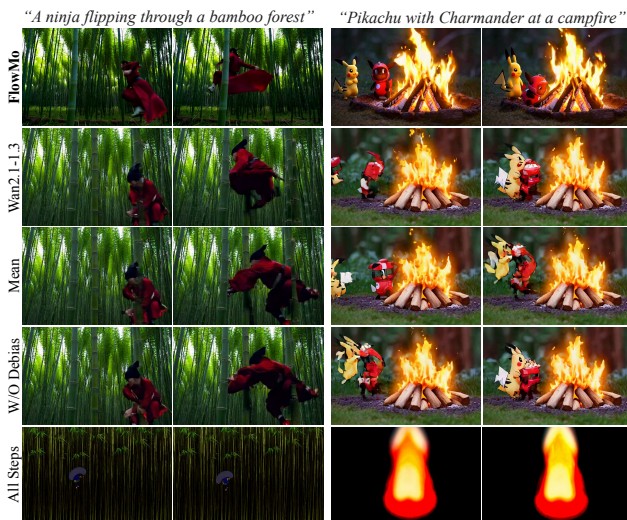

Figure 7: **Ablation study.** We ablate the main design choices of FlowMo, i.e., using the maximal variance for the objective (3rd row), using the appearance-debiasing operator (4th row), the selection of the optimization steps (5th row), and show that FlowMo is significantly superior to all variants.

troduces artifacts, as the optimization interferes with high frequencies and fine details in steps where motion is already determined.

## 4.5 Limitations

While our method enables substantial improvements in motion coherence and overall quality of generated videos, it still has a few limitations. First, due to the calculation and propagation of gradients by our method (Alg. 1), there is some slowdown in the inference time. On average, generating a video with FlowMo takes 160.67 seconds compared to 99.27 seconds without it, corresponding to a $\times 1.82$ increase. This overhead could be mitigated by integrating FlowMo into the training phase, eliminating the need for gradient-based optimization at inference time.

Second, since FlowMo does not modify the model weights, it is bounded by the learned capabilities of the pre-trained model. While it can improve the coherence of motion predicted by the model, it cannot synthesize motion types the model has not learned to represent. We believe this limitation can be addressed by incorporating motion-based objectives *based on the model's internal representations* during training, encouraging richer temporal understanding in generative video models.

## 5 Conclusions

Can we extract meaningful temporal representations from a model with limited temporal understanding? In this work, we propose a new approach to address temporal artifacts in text-to-video models. Instead of relying on external signals, additional data, or specialized architectures, we *repurpose the model's own learned representations as a source of temporal guidance*. Specifically, we find that the semantic latent space learned by text-to-video diffusion models implicitly encodes valuable temporal information. Through extensive analysis (Sec. 3.2), we show that distances between pairs of frames in this latent space correlate with intuitive measures of temporal artifacts, such as patch-wise variance over time. Building on these insights, we implement an inference-time guidance method that encourages smoother transitions in the latent space, and observe that this maps to smoother behavior in pixel space as well, significantly boosting motion coherence while preserving and even improving other aspects of the generation. We hope this work sparks further interest in exploring the temporal properties of semantic latent spaces and encourages the development of methods that improve temporal coherence by looking inward rather than outward.

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

# A    Comparison between FlowMo and FreeInit

In this appendix, we compare FlowMo with FreeInit [59], which is most closely related to our method. FreeInit was designed for earlier UNet-based models that employ DDPM or DDIM [61, 45]. Its approach is motivated by the observation that such models often exhibit significant spatio-temporal inconsistencies in scene elements across frames (e.g., character identities and backgrounds changing between frames). Their primary observation is that these models exhibit discrepancies in signal-to-noise ratios (SNR) between training and inference phases, causing temporal artifacts. In contrast, modern Transformer-based architectures, as used in our work, are generally more robust to these inconsistencies due to more powerful architectures, more stable training frameworks (using FM), and larger training datasets. Thus, these models are able to maintain consistent appearance across frames and are less susceptive to these types of artifacts.

To provide a fair comparison, we adapted FreeInit for use with FM-based DiT models. This involved re-noising a denoised latent and combining this re-noised latent with random noise to initialize the low-frequency components, before repeating the denoising process, as per FreeInit's methodology. We then conducted both quantitative (user study and VBench automatic metrics) and qualitative comparisons between FlowMo and our adapted FreeInit. The experiments are demonstrated hereafter.

**Implementation details.**    All experiments were done on the Wan2.1-1.3B model, for efficiency. The experimental setting for the vanilla baseline and FlowMo-guided model is the same as in Sec. 4. FreeInit was implemented based on its publicly available open-source code [59] and its default configuration, namely employing the `butterworth` filter with $n = 4$, $d_s = d_t = 0.25$. To remain comparable with our work, we performed one refinement iteration with each of the methods.

## A.1    User Study

Consistent with the experiments presented in the main paper, we compare FlowMo to the FreeInit baseline using both the VBench benchmark [16] and human evaluation results.

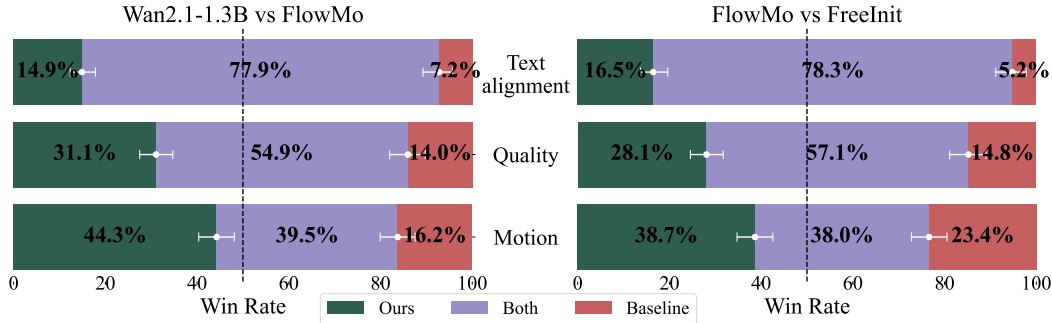

Figure 8: **User studies** conducted on Wan2.1-1.3B [1] using VideoJAM-bench [3]. The studies compare three variants of the model: vanilla (Wan2.1-1.3B), with FlowMo, and with FreeInit. Our method significantly outperforms the baselines in both studies. 95%-confidence intervals were calculated using Dirichlet sampling, assuming a multinomial distribution with Laplace smoothing applied to the counts.

We conducted a human preference study that compared videos generated by Wan2.1-1.3B guided by FlowMo with those guided by FreeInit. Videos from both models were sampled given prompts from the VideoJAM benchmark [3] in the same setting described in Sec. 4.2 of the main paper.

The results, presented in Fig. 8, clearly indicate a strong human preference for FlowMo across all evaluated categories, in comparison to FreeInit as well as the vanilla Wan2.1-1.3B model. Comparing FreeInit and FlowMo, for **Motion Coherence**, FlowMo was preferred in 38.7% of comparisons, substantially more than FreeInit (23.4%). In terms of **Aesthetic Quality**, FlowMo was chosen in 28.1% of pairs, nearly double the preference for FreeInit (14.8%). Furthermore, FlowMo also outperformed FreeInit in **Text-Video Alignment**, with a preference rate of 16.5% compared to FreeInit's 5.2%. These results demonstrate that human evaluators find FlowMo-guided videos to be

significantly more coherent, aesthetically pleasing, and better aligned with textual prompts than those guided by the adapted FreeInit.

## A.2 VBench Benchmark

Table 2: **VBench evaluation results per dimension.** Each column represents a model variant, Wan2.1 (Baseline), with FlowMo, and with FreeInit. Rows correspond to the 16 VBench evaluation dimensions. While FlowMo significantly increases the overall quality of the videos (+6.20%), FreeInit reduces it (-0.08%), and is unable to compare with the Baseline and FlowMo in any of the dimensions.

| Dimension | Wan2.1-1.3B | | |
|---|---|---|---|
| | Baseline | + FlowMo | + FreeInit |
| Subject Consistency | 95.61% | **96.54%** | 93.71% |
| Background Consistency | **97.25%** | 97.02% | 97.12% |
| Temporal Flickering | **99.15%** | 98.93% | 97.77% |
| Motion Smoothness | 96.43% | **98.56%** | 96.54% |
| Dynamic Degree | **83.21%** | 81.96% | 67.01% |
| Aesthetic Quality | 56.77% | **58.03%** | 50.99% |
| Imaging Quality | 61.01% | **64.89%** | 57.19% |
| Object Class | 91.37% | **95.35%** | 92.11% |
| Multiple Objects | 77.98% | **82.27%** | 73.26% |
| Human Action | **98.27%** | 97.23% | 97.98% |
| Color | **87.94%** | 87.28% | 86.53% |
| Spatial Relationship | 75.21% | **78.42%** | 76.52% |
| Scene | **49.84%** | 49.41% | 48.32% |
| Appearance Style | 20.95% | **28.45%** | 21.59% |
| Temporal Style | 26.35% | **27.30%** | 24.24% |
| Overall Consistency | 23.65% | **25.53%** | 22.13% |
| **Semantic Score** | 84.70% | **89.11%** | 85.91% |
| **Quality Score** | 65.58% | **73.58%** | 64.22% |
| **Final Score** | 75.14% | **81.34%** (+6.20%) | 75.06 (-0.08%) |

We further evaluate FlowMo against the adapted FreeInit using the VBench benchmark. The detailed results per dimension are presented in Tab. 2.

The VBench metrics [16] corroborate the user study findings, showing FlowMo's superiority. First, observe that across the comprehensive suite of VBench metrics detailed in Tab. 2, *the adapted FreeInit does not achieve a superior score to both FlowMo and the baseline in **any** individual dimension*.

Considering the main metrics related to motion coherence, FreeInit achieves only a marginal improvement in **Motion Smoothness** (+0.11% over baseline) compared to FlowMo's substantial +2.13% gain. Critically, FreeInit significantly degrades the **Dynamic Degree** by -16.20% from the baseline (from 83.21% to 67.01%), whereas FlowMo maintains a comparable dynamic level (81.96%). This large reduction in motion by FreeInit suggests that its apparent coherence might stem from producing less dynamic videos, which are inherently easier to keep coherent, rather than genuinely improving the quality of complex motion. Furthermore, FreeInit performs worse than both the baseline and FlowMo in several other important quality aspects. For instance, its **Aesthetic Quality** (50.99%) is lower than both baseline (56.77%) and FlowMo (58.03%). Similar trends are observed for other important qualities, e.g. **Temporal Flickering** (FreeInit: 97.77% vs. FlowMo: 98.93%, Baseline: 99.15%), **Imaging Quality** (FreeInit: 57.19% vs. FlowMo: 64.89%), **Appearance Style** (FreeInit: 21.59% vs. FlowMo: 28.45%), **Temporal Style** (FreeInit: 24.24% vs. FlowMo: 27.30%), and **Overall Consistency** (FreeInit: 22.13% vs. FlowMo: 25.53%).

Finally, *FlowMo significantly outperforms FreeInit in the aggregated VBench metrics*. FlowMo achieves a **Final Score** of 81.34%, a +6.20% improvement over the baseline, while *FreeInit scores 75.06%, slightly below the baseline*. Similarly, FlowMo leads in **Quality Score** (73.58% vs. FreeInit's 64.22%) and **Semantic Score** (89.11% vs. FreeInit's 85.91%). These results underscore that FlowMo provides a more effective and well-rounded improvement to video generation quality compared to the adapted FreeInit on modern FM-based DiT architectures.

# B  Additional Qualitative Motivation Results

To complement the qualitative observations in the main paper, we present additional visualizations of the latent-space predictions $\bar{z}_{1,c}$ across timesteps during the generation process. The two figures below show a grid of frames from six different time indices (10–20) and ten representative diffusion steps (0–18), providing a spatio-temporal view of how motion emerges over time within the latent space.

Fig. 9 displays predictions for channel 0, used in the main paper, while Fig. 10 shows results for a randomly selected channel (channel 7). In both cases, we observe that coarse spatial structure appears in early steps, while coarse motion emerges primarily between steps 4 and 8. Although motion is refined in later steps, as seen in timesteps $t = 10$ onward, its coarse features are determined earlier, making the first timesteps the most crucial for coherent motion generation. These patterns reinforce our interpretation that motion is added into the generation during these intermediate steps, which underpins our focus on this range for motion-aware optimization in FlowMo.

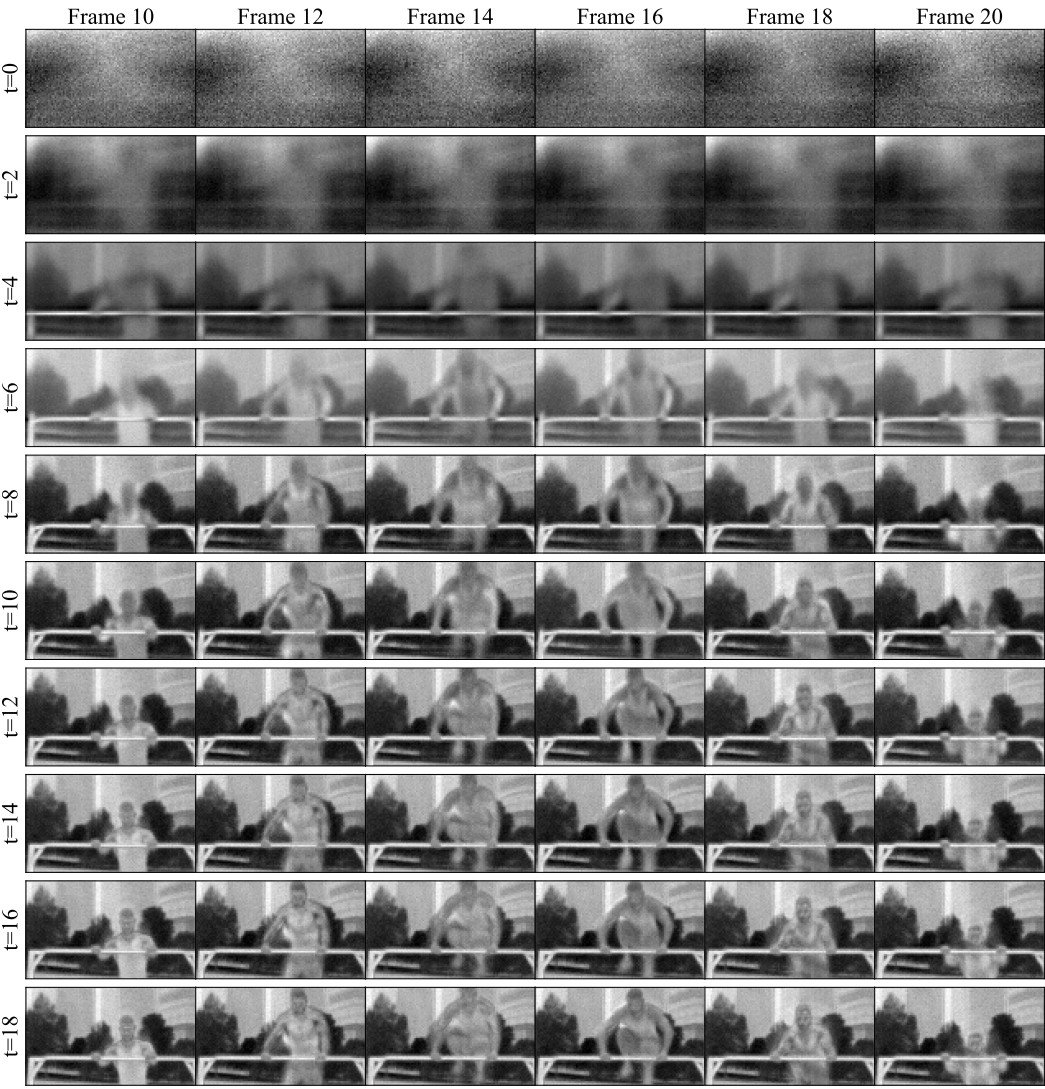

Figure 9: A visualization of channel 0 (selected arbitrarily, and used in the main paper) of the latent prediction at different timesteps of the generation.

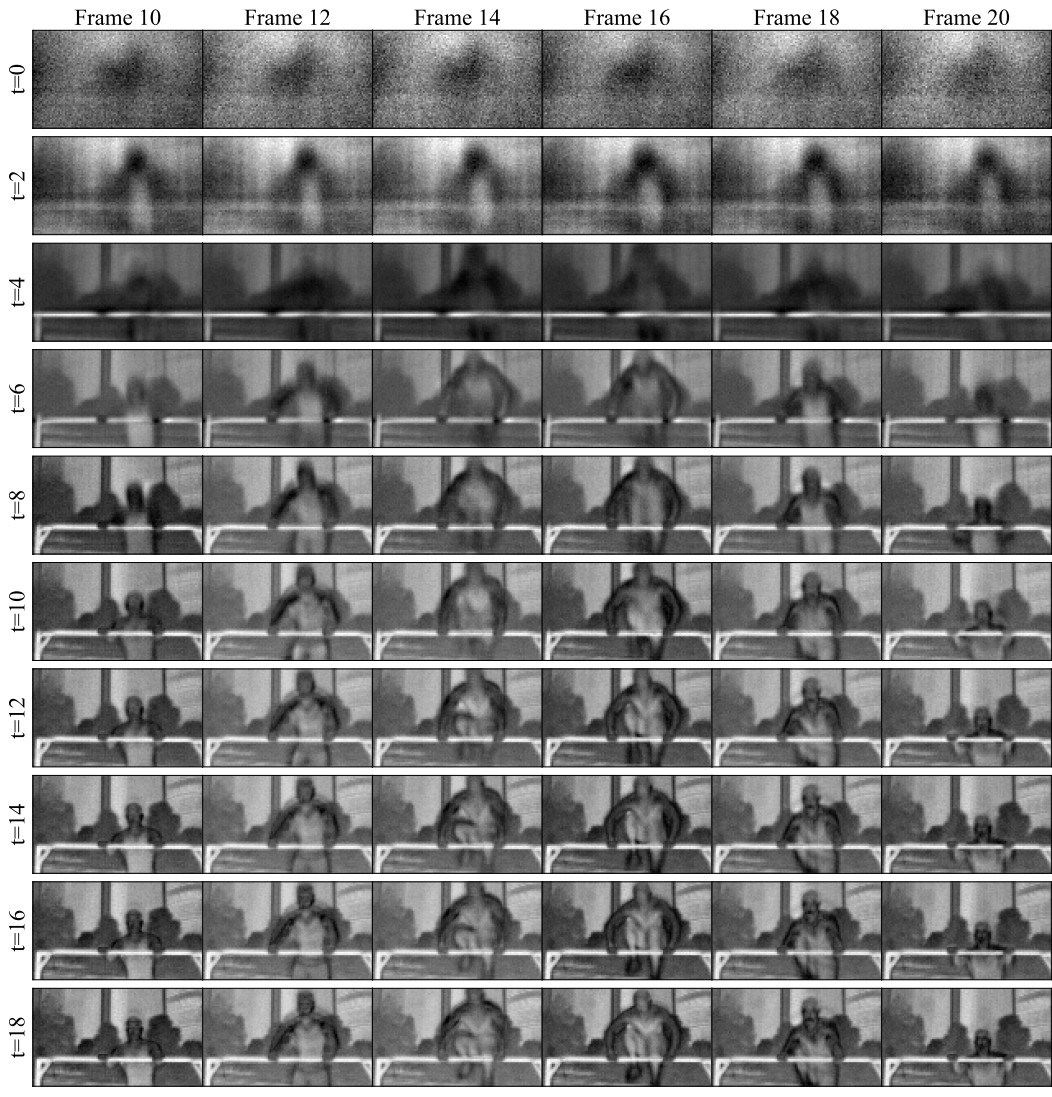

Figure 10: A visualization of channel 7 (selected randomly) of the latent prediction at different timesteps of the generation.

# C   The Effect of FlowMo on Patch-Wise Variance

This section demonstrates the alignment between FlowMo's optimization mechanism and our motivating insights from Sec. 3.2. Fig. 11 plots the maximal patch-wise temporal variance (FlowMo's loss, Equation (7)) for videos generated with and without FlowMo guidance.

We observe that the results demonstrate a strong correlation with the conclusions from Sec. 3.2. First, the application of FlowMo results in a significant reduction and stabilization of this maximal variance, which is particularly evident from approximately timestep 5 onward. This observation is consistent with our earlier finding (Fig. 2) that the variance characteristics of coherent and incoherent videos begin to diverge at this stage.

Second, FlowMo's optimization is applied during the initial 12 timesteps of the generation process. This targeted intervention aligns with our qualitative motivation (Fig. 3), which indicates that coarse motion patterns are predominantly established within these early stages of the generation. As can be observed, applying the optimization at these steps indeed stabilizes the maximal variance in all other, non-optimized steps as well.

Figure 11: **FlowMo effect on patch-wise variance**. We plot the maximal temporal variance of spatial patches for videos with and without applying FlowMo guidance, and observe that our method significantly reduces and stabilizes the variance in the generation steps that impact motion the most by our analysis from Sec. 3.2. 95%-confidence interval was computed using the `seaborn` python package.

Consequently, Fig. 11 illustrates that FlowMo guidance leads to a notable decrease in the maximal patch-wise temporal variance. Videos generated with FlowMo exhibit consistently lower variance, especially within the critical timesteps 5-15, compared to the higher and more fluctuating variance observed in videos generated without FlowMo. This empirically validates that FlowMo operates as intended by reducing the target variance metric during the crucial phases of motion formation.

# D  User Study: Instructions Provided to Participants

As part of the evaluations we performed on our method, we conducted a user study, as described in Sec. 4.2. The study was designed to assess human preferences on videos generated with and without FlowMo, using the videoJAM benchmark [3], which focuses on motion coherence.

The study was conducted using Google Forms. For each prompt, participants were shown a pair of videos—one with FlowMo and one without—generated with the same random seed (1024). The order of the videos was randomized to avoid positional bias. Each pair was evaluated by five different participants, resulting in $640$ responses per baseline.

Participants were asked to evaluate the videos based on three criteria: *text alignment*, *aesthetic quality*, and *motion coherence*. The instructions provided to annotators are reproduced below, followed by a screenshot of the interface used:

---

**Hello!**   We need your help to read a caption, and then watch two generated videos.
After watching the videos, we want you to answer a few questions about them:
- **Text alignment**: Which video better matches the caption?
- **Quality**: Aesthetically, which video is better?
- **Motion**: Which video has more coherent and physically plausible motion?
  *(Do note: it is OK if the quality is less impressive as long as the motion looks better.)*

---

Figure 12: Screenshot of the Google Form used in the user study.

# E    VBench Metrics Breakdown

Table 3: **VBench evaluation results per dimension.** Each column represents a model variant, with and without FlowMo. Rows correspond to the 16 VBench evaluation dimensions.

| Dimension | Wan2.1-1.3B | | CogVideoX-5B | |
|---|---|---|---|---|
| | **Baseline** | **+ FlowMo** | **Baseline** | **+ FlowMo** |
| Subject Consistency | 95.61% | **96.54%** | 95.79% | **97.02%** |
| Background Consistency | **97.25%** | 97.02% | 96.53% | **97.53%** |
| Temporal Flickering | **99.15%** | 98.93% | **99.23%** | 96.21% |
| Motion Smoothness | 96.43% | **98.56%** | 95.01% | **97.29%** |
| Dynamic Degree | **83.21%** | 81.96% | **65.29%** | 63.92% |
| Aesthetic Quality | 56.77% | **58.03%** | 55.51% | **58.29%** |
| Imaging Quality | 61.01% | **64.89%** | **58.91%** | 58.75% |
| Object Class | 91.37% | **95.35%** | 82.72% | **88.41%** |
| Multiple Objects | 77.98% | **82.27%** | 60.17% | **61.27%** |
| Human Action | **98.27%** | 97.23% | **97.81%** | 95.69% |
| Color | **87.94%** | 87.28% | **82.75%** | 80.82% |
| Spatial Relationship | 75.21% | **78.42%** | **67.89%** | 67.23% |
| Scene | **49.84%** | 49.41% | 51.55% | **54.13%** |
| Appearance Style | 20.95% | **28.45%** | 23.53% | **30.29%** |
| Temporal Style | 26.35% | **27.30%** | 25.04% | **31.26%** |
| Overall Consistency | 23.65% | **25.53%** | **26.43%** | 24.28% |
| **Semantic Score** | 84.70% | **89.11%** | **70.03%** | 69.26% |
| **Quality Score** | 65.58% | **73.58%** | 60.83% | **72.11%** |
| **Final Score** | 75.14% | **81.34%** (+6.20%) | 65.43% | **70.69%** (+5.26%) |

In Sec. 4.2, we reported the aggregated VBench [16] metrics, as well as specific metrics that correspond to motion coherence and magnitude. Here, we provide the complete breakdown across all 16 individual evaluation dimensions for both Wan2.1 and CogVideoX.

Tab. 3 compares the baseline models with their FlowMo-guided counterparts. FlowMo leads to consistent improvements in key dimensions, including *Subject Consistency*, *Motion Smoothness*, and *Object Class*, while maintaining or slightly improving aesthetic and perceptual metrics such as *Aesthetic Quality*, *Appearance Style*, and *Spatial Relationship*. Although a small decrease is observed in *Dynamic Degree*, this aligns with our expectation that reducing motion artifacts also reduces spurious motion.

Critically, as mentioned in the main text, FlowMo consistently and significantly boosts the overall quality metric (**Final Score**) by at least 5% across all models. This is a clear indication of the positive impact our method has on the overall quality of the produced viseos.

# F    Additional Quantitative Ablation Results

Table 4: Comparison of baseline(WAN2.1-1.3B), FlowMo, and ablations across VBench metrics. FlowMo achieves the best Motion Smoothness and the highest overall Final Score.

| Dimension | Mean | W/O Debias | Steps 0–30 | Baseline | +FlowMo |
|---|---|---|---|---|---|
| Subject Consistency | 94.26% | 95.02% | 83.56% | 95.61% | **96.54%** |
| Background Consistency | **97.69%** | 96.72% | 92.24% | 97.25% | 97.02% |
| Temporal Flickering | 99.02% | 99.45% | 91.82% | **99.15%** | 98.93% |
| Motion Smoothness | 97.53% | 98.01% | 80.42% | 96.43% | **98.56%** |
| Dynamic Degree | 82.99% | 82.12% | 72.98% | **83.21%** | 81.96% |
| Aesthetic Quality | 53.81% | 54.65% | 43.27% | 56.77% | **58.03%** |
| Imaging Quality | 59.22% | 58.72% | 49.71% | 61.04% | **64.89%** |
| Object Class | 92.89% | 90.74% | 74.33% | 91.37% | **95.34%** |
| Multiple Objects | 78.01% | 78.48% | 58.92% | 77.98% | **82.27%** |
| Human Action | 98.51% | 98.08% | 77.89% | **98.27%** | 97.23% |
| Color | 86.23% | **86.98%** | 82.73% | 87.94% | 87.28% |
| Spatial Relationship | 77.85% | 76.36% | 72.90% | 75.21% | **78.42%** |
| Scene | 47.77% | 48.62% | 40.98% | 49.44% | **49.41%** |
| Appearance Style | 22.03% | 21.22% | 16.04% | 20.95% | **28.45%** |
| Temporal Style | 27.01% | 26.76% | 22.58% | 26.35% | **27.30%** |
| Overall Consistency | 23.81% | 24.92% | 15.87% | 23.65% | **25.53%** |
| Semantic Score | 82.35% | 82.15% | 69.27% | 84.09% | **89.11%** |
| Quality Score | 66.27% | 67.51% | 53.37% | 65.58% | **73.58%** |
| Final Score | 74.76% | 76.31% | 58.17% | 75.14% | **81.34%** |

In Sec. 4.4, we reported the show qualitative results for our ablation study. Here, we provide the complete breakdown across all 16 individual evaluation dimensions for Wan2.1.

Tab. 4 shows a quantitative comparison of VBench metrics between the WAN2.1-1.3B baseline, FlowMo, and the following ablations: **Mean** (i.e., regulating mean patch-wise variance of the debiased latent), **W/O Debias** (i.e., regulating max patch-wise variance of the un-debiased latent), and **Steps 0–30** (applying optimization over the first 30 steps, instead of all 50, due to computational and time constraints).

In accordance with the qualitative results presented in the paper (Fig. 7), FlowMo demonstrates the best Motion Smoothness, with roughly 50% more improvement over the best ablation. It also achieves the strongest performance across the three aggregated metrics: **+5.86%** on Semantic Score, **+6.07%** on Quality Score, and **+5.03%** on Final Score compared to the closest ablation. As for Dynamic Degree, applying optimization over 30 steps leads to a significant drop, whereas the other ablations, and FlowMo itself, show no meaningful degradation, indicating that motion magnitude is largely preserved.

The quantitative metrics are consistent with our motivation and theoretical justification (see Sec. 3.2 and  Appendix E). Replacing the max operation with a mean significantly weakens the effect of the optimization, as was also argued theoretically above. Removing the debiasing operator results in a comparable degradation, which aligns with prior observations that text-to-video models tend to prioritize appearance features, which can overshadow motion in the loss signal (e.g., [3]). Lastly, applying the optimization across the first 30 denoising steps leads to a clear performance drop. This is because the coarse denoising steps primarily determine motion, structure, and overall layout. In later steps, these aspects are already fixed, and the loss, although still valid, can only influence high-frequency details. As a result, applying the optimization at later stages tends to introduce artifacts rather than improve motion quality. This observation aligns with findings from prior works (e.g., [47, 69]), which similarly emphasize the importance of applying optimization during the early denoising stages.

# G   On Choosing to Optimize Maximal Patch-Wise Variance

We provide both intuitive and theoretical justifications for the choice of the max operator in our optimization.

**Intuitive justification.**   Artifacts are typically a local phenomenon dominated by specific spatiotemporal locations in which the patch statistics are abnormal (compared to the training set). Using the mean operator dilutes the impact of these anomalies, as it is dominated by the majority of patches. In contrast, their presence can be captured by a max operator, making it a natural choice to detect and mitigate inconsistencies.

**Theoretical justification.**   The following lemmas justify optimizing the maximum patch-wise variance to keep generated latents in distribution.

**Lemma G.1.** *Noising of real videos keeps patch-wise temporal variance bounded by a dataset-dependent constant $C$, which bounds patch-wise variance for all patches of all videos in the training data.*

Since the model was trained on a finite dataset of real-valued latents, such a constant $C$ exists. Assuming the dataset consists of natural videos, and since VAEs minimize latent variance, $C$ is likely to be relatively small. Recall that noising a latent of a real video $x_0$ to timestep $t$ is done by:

$$x_t = \sqrt{\bar{\alpha}_t} x_0 + \sqrt{1 - \bar{\alpha}_t}\, \epsilon, \quad \epsilon \sim \mathcal{N}(0, I). \tag{10}$$

Thus, the temporal variance of each patch $p$ is given by:

$$\text{Var}_t(p) = (1 - \bar{\alpha}_t)\, \text{Var}(\epsilon_p), \tag{11}$$

and hence the maximum patch-wise variance of all samples during training is bounded.

**Lemma G.2.** *Consider the denoising process starting from pure Gaussian noise. Suppose that in early timesteps (i.e., when $t$ is small), each patch of the denoised latent remains approximately Gaussian but with unknown variance due to guidance, decoder mismatch, or score error, i.e., $x_t^p \sim \mathcal{N}(0, \sigma_p^2)$, with $\sigma_p$ varying across space. Then, the maximal patch-wise variance is $\mathcal{O}(\log(N_f N_w N_h))$ with high probability (w.h.p.), where $N_f$, $N_w$, and $N_h$ are the number of frames, width, and height of the latent, respectively.*

Empirically, this assumption holds in early denoising timesteps: the denoised latent remains approximately Gaussian, and patch-wise variances exhibit spatial variability consistent with the stated form. Intuitively, this holds because at high noise levels, the per-pixel variance distribution is heavy-tailed, so outliers are likely to occur. Formally, given the assumptions above, the temporal variance of each patch is distributed $\chi_{N_f}^2$, where $N_f$ is the number of latent frames. The maximal patch-wise variance is thus the maximum over such $N = N_f N_w N_h$ distributions, which lies in the Gumbel domain and satisfies the property in the lemma.

Hence, as the number of patches $N$ grows, local outliers (i.e., patches with high patch-wise variance) appear with non-negligible probability.

**Lemma G.3.** *A global stability condition for the numerical integration step in the denoising process requires that $|J_p|\, v_p \leq \mathcal{C}$, where $J_p$ is the Jacobian at patch $p$, $v_p$ its patch-wise variance, and $\mathcal{C}$ the Courant number.*

Let $f(x_t)$ denote a linearization of the denoising step. A stability condition for solving the ODE numerically is the CFL condition, which requires every patch $p$ to satisfy $|J_p|\Delta t \leq \mathcal{C}$. Otherwise, the numerical process could be unstable or fail to converge. If the CFL condition holds, the local step magnitude satisfies:

$$\|\Delta x_p\| \approx |J_p|\, v_p, \tag{12}$$

leading to the stated global stability condition. If a single patch attains a large $v_p$, the product above can cross the stability boundary, causing a local overshoot that pushes that region off the training manifold.

**Corollary G.4.** *During training, patch-wise variance is bounded. However, when denoising pure Gaussian noise, high patch-wise variance is a likely local phenomenon, which may drive generation out-of-distribution. Minimizing the maximum patch-wise variance is thus a practical strategy to improve ODE stability and ensure generated latents remain in-distribution.*

