# OpenReview forum: "FlowMo: Variance-Based Flow Guidance for Coherent Motion in Video Generation"
_NeurIPS.cc/2025/Conference — NeurIPS 2025 poster_

### Official Review · Reviewer_tGek · 2025-06-24

**Clarity:** 3
**Significance:** 3
**Originality:** 3
**Rating:** 5
**Confidence:** 3

**Summary:**

This paper presents a training free method for producing coherent videos from a pretrained video generation flow matching model. The idea is to introduce a refinement optimization when sampling the denoising steps that encourages more coherent frame to frame motions.

**Questions:**

The overall idea of the work is simple and nice. But the core idea of adding a refinement optimization based on frame to frame variance in patches seems like it should be tested not just at inference time, but also in the context of training. I would enourage the authors to include in the rebuttal some results for incorporating it into training. How good are the resulting videos in terms of coherence? What is the inference runtime in this setting?

**Ethical Concerns:**

["NO or VERY MINOR ethics concerns only"]

**Final Justification:**

Based on the strengths outlined above and the rebuttal I recommend acceptance.

**Limitations:**

Yes

**Paper Formatting Concerns:**

None.

**Quality:**

3

**Strengths And Weaknesses:**

Strengths:

* Training-free
* The paper does a nice job of showing why a patch-based variance reduction approach can be an effective refinement strategy. I appeciate the experiment described in Section 3.2.

Weaknesses:

* The refinement optimization introduces a 2.39x increase in runtime to perform the gradient descent step. This is very briefly mentioned in the limitations section of the paper. I think this needs to be brought up as part of the results and evaluation. The authors mention that this optimization could also be made more efficient by incorporating it into training. I think the work would be stronger if it showed how adding it to training improves results.

---

> ### Author Rebuttal · Authors · 2025-07-30
>
> We thank the reviewer for their recognition of the training-free design and the effectiveness of our patch-based variance reduction approach.
>
> ## Runtime speedup
>
> In response to the reviewer’s feedback, we made small implementation changes that preserve correctness. Specifically, we replaced redundant operations and removed unnecessary computation (e.g., replacing loops over tensors with built-in PyTorch functions). The code changes did not change the results – we verified on all samples from our project’s website that the obtained outputs are bit-exact identical before and after the change. These optimizations to a x1.29 speedup, reducing the mean runtime from 234.30 seconds to 180.67 seconds. FlowMo now incurs an overhead of only x1.82 (down from x2.39) compared to baseline.
>
> ## Adding FlowMo during training.
>
> Thank you for the insightful suggestion. First, we note that integrating our refinement into training is a promising direction for future research, since prior works like [1] show that inference-time objectives (e.g., [2]) can be effectively adapted into training, and obtain similar gains. Since FlowMo is similar in spirit to these image-based inference-time optimizations, we believe it will yield similar benefits when applied to pretraining or fine tuning.
> However, attempting this is infeasible under the limits of our current computational resources; we only have access to two H100 GPUs, and backpropagating through larger batches with latent gradients would be significantly more demanding than our inference-time setup.
>
> We appreciate your time and constructive feedback. We would be happy to address any further questions or concerns you may have during the discussion period.
>
>
> [1] Separate-and-Enhance: Compositional Finetuning for Text2Image Diffusion Models
>
> [2] Attend-and-Excite: Attention-Based Semantic Guidance for Text-to-Image Diffusion Models

---

> > ### Comment · Reviewer_tGek · 2025-08-05
> >
> > Thank you for the rebuttal. It has addressed my concerns.

---

### Official Review · Reviewer_mc6Y · 2025-07-02

**Clarity:** 3
**Significance:** 1
**Originality:** 2
**Rating:** 4
**Confidence:** 5

**Summary:**

The paper focuses on enhancing the temporal coherence of text-to-video diffusion models. The authors introduce FlowMo, a novel training-free guidance method that aims to improve motion coherence without additional training or external inputs. The main contributions include the Appearance-Debiased Temporal Representation to measure the distances between consecutive frame latents, and Motion Coherence Estimation to guide the model towards reducing temporal inconsistencies during sampling. Experiments conducted on VBench and VideoJAM-bench show reasonable improvements over the baselines without FlowMo.

**Questions:**

1. The questions in weaknesses part.
2. How well does FlowMo perform on models with different architectures or on tasks requiring more nuanced temporal reasoning?
3. Could external factors, such as the quality or specificity of the input text, affect the performance of FlowMo in generating temporally coherent videos?

**Ethical Concerns:**

["NO or VERY MINOR ethics concerns only"]

**Final Justification:**

Thank you for your response. Some of my concerns have been well-addressed, particularly regarding the dynamic degree and ablation studies, which will lead me to consider increasing my rating.
Furthermore, I still highly recommend conducting experiments with a higher-performing implementation of the WAN baseline. The significant performance gap between the officially-released performance and the performance measured in this paper could weaken the paper's contribution.
Finally, regarding the authors' statement in their response, "our experiments were carefully designed to cover all types of motion," I find this claim to be too absolute and suggest toning down the description in this regard.

**Limitations:**

yes

**Quality:**

2

**Strengths And Weaknesses:**

Strengths:
+ Tested across various models like Wan2.1-1.3B and CogVideoX-5B to showcase the method's generalizability.
+ The method does not require retraining or external conditioning signals, making it a versatile solution for improving existing models.
+ The paper is mostly clear and easy to follow.

Weaknesses:
- The technical contribution is limited. As a training-free method, the main idea of FlowMo is to improve the temporal consistency by explicitly reducing the difference between consecutive frames. In this way, this method is more like a trade-off between temporal consistency and dynamic degree. It is evidenced by the degraded Dynamic Degree metric of FlowMo on VBench. Therefore, the technical contribution of this paper is limited.
- The ablation study is needed. The proposed FlowMo is a manually designed rule to refine diffusion model prediction. There are many manually-determined designs in FlowMo including the refinement steps, the calculation of patch-wise variance  and FlowMo loss. The ablation study of how these designs influence the performances should be included.
- The officially reported total score of Wan2.1-1.3B on VBench is 84.26%. However, in this paper, the performance of Wan2.1-1.3B is improved from 75.14% (without FlowMo) to 81.34% (with FlowMo). I speculate that this large gap may be caused by different inference settings (e.g., output resolution, prompt refinement). Since the performance of Wan2.1-1.3B baseline is much lower the official version, the conclusion in this paper is not convincing enough.
- The results might be influenced by external factors such as the complexity of the input text prompts or the diversity of the generated content. For the prompt that requires large motion or complex transition, the continuous constrain may affect the results.
- The longer inference time than baseline may limit the application of FlowMo on larger model.

---

> ### Author Rebuttal · Authors · 2025-07-30
>
> We thank the reviewer for highlighting the method’s generalizability across models and its versatility as a training-free, plug-and-play refinement.
>
> ## Trade-off between temporal consistency and dynamic degree
>
> Please note that this is addressed directly in both the motivational experiments and the main results of the paper, showing that FlowMo improves temporal coherence without meaningfully compromising motion dynamics.
>  1. In our motivation (Sec. 3.2, L. 144–152), we specifically analyze high-motion videos and find that patch-wise variance correlates with coherence even when motion is high, indicating our loss targets coherence, not reduced dynamics.
>  2. In our main results (Sec. 4.2, App. A.2, Table 2), we show that FlowMo reduces Dynamic Degree by less than 1.5% across both models, a negligible drop given the overwhelming benefit to the overall Quality Score (Wan2.1: +8.0%, CogVideoX: +11.25%). In contrast, FlowMo’s closest baseline, FreeInit, causes a 16.2% drop in Dynamic Degree (see L. 568-574, Appendix A). This is a further indication that our method does not incur a significant drop in dynamic degree, even compared to its closest baselines.
>  3. Finally, note that Dynamic Degree reflects motion *magnitude*, not *coherence*; for instance, completely incoherent motion (e.g. flickering) can inflate Dynamic Degree scores. Thus, slight reductions in this score are expected when coherence improves.
>
> Therefore, rather than exploiting a trade-off between the two measures, FlowMo improves coherence without compromising dynamic degree.
>
> To provide a clear overview, we briefly summarize the core contributions of our work:
>  - We present a systematic analysis of motion in video diffusion models (Sec. 3.2), revealing that motion often emerges in later denoising steps, distinct from appearance features. We justify our insights both quantitatively and qualitatively. We further show that patch-wise variance is a strong predictor of temporal coherence, even in high-motion videos (Fig. 2, L. 144-154).
>  - We propose FlowMo, a training-free, inference-time method that significantly improves temporal coherence *with minimal impact on motion dynamics*. It is lightweight, plug-and-play, and requires no retraining.
>  - We demonstrate strong cross-model generalization across Wan2.1 and CogVideoX (which differ in architecture, training data, and base performance) highlighting the robustness and broad applicability of our approach.
>
> ## Justification of design choices.
>
> Due to the character limit, we are unable to include the full response here. We kindly refer the reviewer to our response to Reviewer gJqQ, where we provide a theoretical justification for using the maximum operator in FlowMo’s loss. That response also includes both empirical and theoretical justifications for the choice of the 1–12 timestep range.
>
> Furthermore, the table below shows a quantitative comparison of VBench metrics between the WAN2.1-1.3B baseline, FlowMo, and the following ablations: Mean (i.e., regulating mean patch-wise variance of the debiased latent, as opposed to max), W/O Debias (i.e., regulating max patch-wise variance of the un-debiased latent), and Steps 0-30 (applying optimization over the first 30 steps, instead of all 50, due to computational and time constraints).
>
> In accordance with the qualitative results presented in the paper (Fig. 6), FlowMo demonstrates the best Motion Smoothness, with roughly 50% more improvement over the best ablation. It also achieves the strongest performance across the three aggregated metrics: +5.86% on Semantic Score, +6.07% on Quality Score, and +5.03% on Final Score compared to the closest ablation. As for Dynamic Degree, applying optimization over 30 steps leads to a significant drop, whereas the other ablations—and FlowMo itself—show no meaningful degradation, indicating that motion magnitude is largely preserved.
>
> The quantitative metrics are consistent with our motivation and theoretical justification. First, replacing the maximum operation with a mean significantly weakens the effect of the optimization, as was also argued theoretically above. Removing the debiasing operator results in a comparable degradation, which aligns with prior observations that text-to-video models tend to prioritize appearance features, which can overshadow motion in the loss signal (e.g., [2]). Lastly, applying the optimization across the first 30 denoising steps leads to a clear performance drop. This is because the coarse denoising steps primarily determine motion, structure, and overall layout. In later steps, these aspects are already fixed, and the loss, although still valid, can only influence high-frequency details. As a result, applying the optimization at later stages tends to introduce artifacts rather than improve motion quality. This observation aligns with findings from prior works (e.g., [1], [3]), which similarly emphasize the importance of applying optimization during the early denoising stages.
>
> | Dimension             | Mean   | W/O Debias | Steps 0-30 | Baseline | + FlowMo |
> |----------------------|--------|------------|-----------|----------|----------|
> | Subject Consistency  | 94.26% | 95.02%     | 83.56%    | 95.61%   | **96.54%**   |
> | Background Consistency | **97.69%** | 96.72%     | 92.24%    | 97.25%   | 97.02%   |
> | Temporal Flickering  | 99.02% | 99.45%     | 91.82%    | **99.15%**   | 98.93%   |
> | Motion Smoothness    | 97.53% | 98.01%     | 80.42%    | 96.43%   | **98.56%**   |
> | Dynamic Degree       | 82.99% | 82.12%     | 72.98%    | **83.21%**   | 81.96%   |
> | Aesthetic Quality    | 53.81% | 54.65%     | 43.27%    | 56.77%   | **58.03%**   |
> | Imaging Quality      | 59.22% | 58.72%     | 49.71%    | 61.01%   | **64.89%**   |
> | Object Class         | 92.89% | 90.97%     | 74.33%    | 91.37%   | **95.35%**   |
> | Multiple Objects     | 78.01% | 78.48%     | 58.92%    | 77.98%   | **82.27%**   |
> | Human Action         | 98.51% | 98.08%     | 77.89%    | 98.27%   | **97.23%**   |
> | Color                | 86.23% | 86.98%     | 82.73%    | **87.94%**   | 87.28%   |
> | Spatial Relationship | 77.85% | 76.36%     | 72.09%    | 75.21%   | **78.42%**   |
> | Scene                | 47.77% | 48.62%     | 40.98%    | **49.84%**   | 49.41%   |
> | Appearance Style     | 22.03% | 21.22%     | 16.04%    | 20.95%   | **28.45%**   |
> | Temporal Style       | 27.01% | 26.76%     | 22.58%    | 26.35%   | **27.30%**   |
> | Overall Consistency  | 23.81% | 24.92%     | 15.87%    | 23.65%   | **25.53%**   |
> | **Semantic Score**   | 83.25% | 85.12%     | 62.97%    | 84.70%   | **89.11%**   |
> | **Quality Score**    | 66.27% | 67.51%     | 53.37%    | 65.58%   | **73.58%**   |
> | **Final Score**      | 74.76% | 76.31%     | 58.17%    | 75.14%   | **81.34%**   |
>
> ## VBench results on Wan2.1
>
> While the official VBench score for Wan2.1-1.3B is reported as 84.26%, the exact inference settings (e.g., resolution, prompt formulation) used to obtain that result have not been publicly detailed. We suspect the performance gap is due to our use of 480p resolution, which is lower than the model’s maximum supported resolution (720p, as stated on the official website). Due to computational limitations, we were unable to run our experiments at the higher resolution. Please note that all our experiments were conducted using the publicly released code and checkpoints, and we followed the default configurations as provided in both the Wan2.1 and VBench repositories. Our goal was to ensure consistency and reproducibility, and under these conditions, FlowMo demonstrates clear and meaningful improvements.
>
> ## Influence by external factors.
>
> We would like to kindly emphasize that our experiments were carefully designed to cover all types of motion, with various complexities, prompts, and settings. Firstly, the VBench dataset includes a wide variety of prompts, ranging from simple to complex, and from static scenes to high-motion scenarios. Secondly, we enclose studies conducted on VideoJAM-bench [2] (Fig. 5,8, L. 247-253), which was carefully and specifically designed to examine motion, and includes highly complex motion types such as gymnastics, physics, rotations, etc. Across these diverse and comprehensive benchmarks, FlowMo consistently shows strong performance and improved temporal coherence, further supporting its robustness to prompt complexity and motion diversity while improving temporal consistency.
>
> ## Runtime speedup
>
> In response to the reviewer’s feedback, we made small implementation changes that preserve correctness. Specifically, we replaced redundant operations and removed unnecessary computation (e.g., replacing loops over tensors with built-in PyTorch functions). The code changes did not change the results – we verified on all samples from our project’s website that the obtained outputs are bit-exact identical before and after the change. These optimizations led to a x1.29 speedup, reducing the mean runtime from 234.30s to 180.67s. As a result, FlowMo now incurs an overhead of only x1.82 (down from x2.13) relative to the baseline.
>
> ---
>
> Thank you once again for your helpful observations and careful review. We would be pleased to further clarify any remaining points or engage in additional discussion.
>
>
> **References**
>
> [1] Attend-and-Excite: Attention-Based Semantic Guidance for Text-to-Image Diffusion Models
>
> [2] VideoJAM: Joint Appearance-Motion Representations for Enhanced Motion Generation in Video Models
>
> [3] Image Generation from Contextually-Contradictory Prompts

---

> > ### Comment · Reviewer_mc6Y · 2025-08-07
> >
> > Thank you for your response. Some of my concerns have been well-addressed, particularly regarding the dynamic degree and ablation studies, which will lead me to consider increasing my rating.
> > Furthermore, I still highly recommend conducting experiments with a higher-performing implementation of the WAN baseline. The significant performance gap between the officially-released performance and the performance measured in this paper could weaken the paper's contribution.
> > Finally, regarding the authors' statement in their response, "our experiments were carefully designed to cover all types of motion," I find this claim to be too absolute and suggest toning down the description in this regard.

---

> ### Author Response · Authors · 2025-08-07
>
> Thank you for your continued engagement and constructive feedback. We are pleased that our responses have addressed several of your concerns, particularly regarding dynamic degree and ablation studies.
>
> ## **Regarding the WAN2.1 baseline performance gap:**
> We appreciate your recommendation to conduct experiments with higher-performing WAN baseline implementations. We want to clarify an important technical constraint we encountered following this discussion: when attempting to run WAN2.1-T2V-1.3B with the flag `--size 1280*720` to match the reported official resolution, we encounter an assertion error:
>
> > **"AssertionError: Unsupport size 1280\*720 for task t2v-1.3B, supported sizes are: 480\*832, 832\*480."**
>
> This technical limitation is also documented in the official WAN2.1 GitHub repository, which states:
>
> > **"The 1.3B model is capable of generating videos at 720P resolution. However, due to limited training at this resolution, the results are generally less stable compared to 480P."**
>
> Additionally, the HuggingFace model card confirms that this specific resolution is not supported in the published release.
>
> Given these constraints, our experiments were conducted at the maximum practically supported resolution (480*832) using the publicly available codebase. While this may contribute to the performance gap you noted, it ensures our results are reproducible and based on the actual capabilities of the released model. Importantly, the benchmark motion metrics that we report are similar to the official WAN2.1 motion metrics, and these are the most relevant to our paper's focus on temporal coherence improvements. Additionally, we would like to highlight that FlowMo improves motion-related metrics even when compared to the published results obtained at higher resolution. Specifically, we achieve:
>
> - **98.56% vs 97.44%** in **Motion Smoothness**
> - **27.30% vs 25.32%** in **Temporal Style**
>
> Our method's effectiveness is demonstrated consistently across both WAN2.1 and CogVideoX models, with the latter showing even stronger improvements (+11.25% vs +8.0% Quality Score improvement), suggesting our approach's robustness extends beyond any single baseline implementation.
>
> ## **Regarding motion coverage:**
> We acknowledge and accept your feedback that our statement "our experiments were carefully designed to cover all types of motion" in the rebuttal was overly absolute. Since the rebuttal cannot be edited at this stage, we recognize this phrasing and agree it should be more measured. A more accurate characterization would be that our experiments cover a diverse range of motion types across comprehensive benchmarks (VBench and VideoJAM-bench), demonstrating consistent improvements across various motion complexities and scenarios.
>
> We believe these clarifications, combined with our previous responses on dynamic degree preservation and ablation studies, strengthen the paper's contribution while maintaining transparency about technical limitations and scope.

---

### Official Review · Reviewer_MYjg · 2025-07-02

**Clarity:** 3
**Significance:** 3
**Originality:** 3
**Rating:** 4
**Confidence:** 4

**Summary:**

This paper aims to enhance motion coherence of text-to-video diffusion models in a training-free way. Text-to-video diffusion models often struggle to accurately capture temporal dynamics like motion and interactions. Existing solutions require retraining or external signals to improve temporal consistency, which can be resource-intensive and complex. FlowMo introduces a training-free guidance approach that leverages the model’s own predictions at each diffusion step to extract appearance-debiased temporal representations and dynamically guide the model to enhance motion coherence by reducing temporal variance. Comprehensive experiments on various text-to-video models show that FlowMo effectively boosts motion coherence without compromising visual quality or prompt alignment, providing an easy-to-integrate improvement for existing models.

**Questions:**

Suggestions:
1. Discuss the cost-effectiveness of FlowMo and its proper applications.
2. Compare FreeInit quantitatively and qualitatively in the main text. And it would be better if more training-free works could be taken into comparison.
3. Add quantitative ablation results to make the paper claims solid.
4. Conduct a careful writing check.

**Ethical Concerns:**

["NO or VERY MINOR ethics concerns only"]

**Final Justification:**

Most of my concerns are addressed in the author's rebuttal except the runtime problem. I appreciate the authors' efforts to speed up the algorithm to fit it in real-world applications, but 1.82x runtime is still hard to be ignored, especially for a method which is claimed to be training-free.
Given the novelties and practical impacts of this work, I would keep my initial score for this work.

**Limitations:**

Yes

**Quality:**

3

**Strengths And Weaknesses:**

Strengths:
1. The quantitative and qualitative motivation analyses are interesting and convincing. The demos in webpage make it clear for readers to understand how the proposed method works.
2. The proposed method is simple and easy to be implemented. The training-free design allows readers to validate the effectiveness of this method easily and inexpensively on their own tasks.
3. The writing is easy to understand.

Weaknesses
1. Computation time is increased by x2.39 when using FlowMo, which will undermine the cost-effectiveness of the method in practical applications.
2. The quantitative and qualitative comparisons with FreeInit should be conduct and displayed in the main text, since it is similar and related to this paper. Instead, this paper majorly compares FreeInit in appendix, which is not very clear for readers.
3. The ablation study chapter is not very clear and precise, since there are no quantitative ablation results.
4. There are some confusing statements. For instance, in terms of the line.3 of Algorithm.1, the first term of the right equation is written as conditioned on P, which is inconsistent with the definition of the CFG formula.

---

> ### Author Rebuttal · Authors · 2025-07-30
>
> We thank the reviewer for the encouraging feedback and are glad they found the motivation analysis convincing, the method simple and accessible, and the writing clear.
>
> ## Runtime speedup
>
> In response to the reviewer’s feedback, we made small implementation changes that preserve correctness. Specifically, we replaced redundant operations and removed unnecessary computation (e.g., replacing loops over tensors with built-in PyTorch functions). The code changes did not change the results – we verified on all samples from our project’s website that the obtained outputs are bit-exact identical before and after the change. These optimizations led to a x1.29 speedup, reducing the mean runtime from 234.30s to 180.67s. As a result, FlowMo now incurs an overhead of only x1.82 (down from x2.13) relative to the baseline.
>
> ## Moving FreeInit to the main text.
>
> Thank you. Following your suggestion, we will move both the quantitative and qualitative comparisons of FreeInit from the appendix into the main text.
>
> ## Additional quantitative ablation results.
>
> The table below shows a quantitative comparison of VBench metrics between the WAN2.1-1.3B baseline, FlowMo, and the following ablations: Mean (i.e., regulating mean patch-wise variance of the debiased latent, as opposed to max), W/O Debias (i.e., regulating max patch-wise variance of the un-debiased latent), and steps 0-30 (applying optimization over the first 30 steps, instead of all 50, due to computational and time constraints).
>
> In accordance with the qualitative results presented in the paper (Fig. 6), FlowMo demonstrates the best Motion Smoothness, with roughly 50% more improvement over the best ablation. It also achieves the strongest performance across the three aggregated metrics: +5.86% on Semantic Score, +6.07% on Quality Score, and +5.03% on Final Score compared to the closest ablation. As for Dynamic Degree, applying optimization over 30 steps leads to a significant drop, whereas the other ablations—and FlowMo itself—show no meaningful degradation, indicating that motion magnitude is largely preserved.
>
> The quantitative metrics are consistent with our motivation and theoretical justification. First, replacing the maximum operation with a mean significantly weakens the effect of the optimization, as was also argued theoretically above. Removing the debiasing operator results in a comparable degradation, which aligns with prior observations that text-to-video models tend to prioritize appearance features, which can overshadow motion in the loss signal (e.g., [4]). Lastly, applying the optimization across the first 30 denoising steps leads to a clear performance drop. This is because the coarse denoising steps primarily determine motion, structure, and overall layout. In later steps, these aspects are already fixed, and the loss, although still valid, can only influence high-frequency details. As a result, applying the optimization at later stages tends to introduce artifacts rather than improve motion quality. This observation aligns with findings from prior works (e.g., [1], [3]), which similarly emphasize the importance of applying optimization during the early denoising stages.
>
> | Dimension             | Mean   | W/O Debias | Steps 0-30 | Baseline | + FlowMo |
> |----------------------|--------|------------|-----------|----------|----------|
> | Subject Consistency  | 94.26% | 95.02%     | 83.56%    | 95.61%   | **96.54%**   |
> | Background Consistency | **97.69%** | 96.72%     | 92.24%    | 97.25%   | 97.02%   |
> | Temporal Flickering  | 99.02% | 99.45%     | 91.82%    | **99.15%**   | 98.93%   |
> | Motion Smoothness    | 97.53% | 98.01%     | 80.42%    | 96.43%   | **98.56%**   |
> | Dynamic Degree       | 82.99% | 82.12%     | 72.98%    | **83.21%**   | 81.96%   |
> | Aesthetic Quality    | 53.81% | 54.65%     | 43.27%    | 56.77%   | **58.03%**   |
> | Imaging Quality      | 59.22% | 58.72%     | 49.71%    | 61.01%   | **64.89%**   |
> | Object Class         | 92.89% | 90.97%     | 74.33%    | 91.37%   | **95.35%**   |
> | Multiple Objects     | 78.01% | 78.48%     | 58.92%    | 77.98%   | **82.27%**   |
> | Human Action         | 98.51% | 98.08%     | 77.89%    | 98.27%   | **97.23%**   |
> | Color                | 86.23% | 86.98%     | 82.73%    | **87.94%**   | 87.28%   |
> | Spatial Relationship | 77.85% | 76.36%     | 72.09%    | 75.21%   | **78.42%**   |
> | Scene                | 47.77% | 48.62%     | 40.98%    | **49.84%**   | 49.41%   |
> | Appearance Style     | 22.03% | 21.22%     | 16.04%    | 20.95%   | **28.45%**   |
> | Temporal Style       | 27.01% | 26.76%     | 22.58%    | 26.35%   | **27.30%**   |
> | Overall Consistency  | 23.81% | 24.92%     | 15.87%    | 23.65%   | **25.53%**   |
> | **Semantic Score**   | 83.25% | 85.12%     | 62.97%    | 84.70%   | **89.11%**   |
> | **Quality Score**    | 66.27% | 67.51%     | 53.37%    | 65.58%   | **73.58%**   |
> | **Final Score**      | 74.76% | 76.31%     | 58.17%    | 75.14%   | **81.34%**   |
>
> ## Inconsistency with CFG Definition.
>
> Thank you for pointing this out. We will revise line 3 In Algorithm 1 to match the correct CFG structure:
> $ u_{\theta, t_i} \gets u_{\theta, t_i \mid \emptyset} + \rho \cdot (u_{\theta, t_i \mid P} - u_{\theta, t_i \mid \emptyset} ) $
>
> ## Cost-effectiveness of FlowMo and its proper applications.
>
> FlowMo is designed to be cost-effective: it requires no additional training and introduces only a modest runtime overhead (which is now x1.82) during inference. This makes it practical for applications where motion quality is critical, such as content creation or video refinement, while preserving efficiency.
>
> ## Additional training-free baselines
>
> In addition to the training-free methods that we have included in our study, the only relevant method that we could not include is VideoGuide. This is due to the incompatible architecture, since this method was developed for a U-net based architecture and is not easily adaptable (see lines 96-103).
>
> We are happy to conduct further comparisons upon request.
>
> ## Conduct a careful writing check.
>
> We will conduct a thorough review of the writing to improve clarity and resolve any ambiguities in the final version of the paper.
>
> ---
>
> We sincerely thank you for your insights and feedback. We would be glad to answer any follow-up questions and continue the discussion.
>
>
> **References**
>
> [1] Attend-and-Excite: Attention-Based Semantic Guidance for Text-to-Image Diffusion Models
>
> [2] FreeInit: Bridging Initialization Gap in Video Diffusion Models
>
> [3] Image Generation from Contextually-Contradictory Prompts
>
> [4] VideoJAM: Joint Appearance-Motion Representations for Enhanced Motion Generation in Video Models

---

> > ### Comment · Reviewer_MYjg · 2025-08-01
> >
> > Thanks for your rebuttal. Most of my concerns are addressed in the author's rebuttal except the runtime problem.
> >
> >  I appreciate the authors' efforts to speed up the algorithm to fit it in real-world applications, but 1.82x runtime is still hard to be ignored, especially for a method which is claimed to be training-free.

---

> > > ### Author Response · Authors · 2025-08-01
> > >
> > > We sincerely thank the reviewer for taking the time to engage in this discussion and for the swift reply. We are pleased that the rebuttal addressed most concerns and are grateful for the opportunity to clarify the remaining point regarding runtime.
> > >
> > > We would like to provide some additional context and relevant comparisons regarding the runtime overhead of FlowMo. Kindly note that all inference-time optimization methods we are aware of incur comparable or greater slowdowns. Two particularly impactful examples include:
> > > 1. Attend-and-Excite [1], which introduces semantic attention guidance during inference and results in a ×1.73 slowdown. Despite this, the method has been cited nearly 600 times, reflecting both the relevance of the challenge it addresses and the broad interest in inference-time optimization.
> > >
> > > 2. ConsiStory [2], a training-free approach for improving subject consistency across varying prompts, reports a runtime of 24 seconds per image, compared to 12 seconds for its baseline, a ×2 slowdown.
> > >
> > > In this context, we believe FlowMo remains highly competitive. Our optimized implementation now runs at ×1.82 overhead (down from ×2.39), using only simple engineering improvements that do not affect output fidelity.
> > >
> > > Additionally, as detailed in our response to Reviewer gJqQ (“Adaptive Timestep Selection”), we explored an adaptive variant of FlowMo that dynamically selects which timesteps to optimize. This approach further reduces runtime to ×1.58, with only a slight decrease in performance relative to the fixed 12-step version. This flexibility allows FlowMo to be adapted to settings where speed is more critical.
> > >
> > > Finally, we reiterate that temporal inconsistency remains one of the most significant challenges in video generation, even in state-of-the-art models. FlowMo addresses this issue in a training-free and model-agnostic way, and we believe that its core insights may help guide the development of future training-time methods that preserve performance without runtime overhead.
> > >
> > >
> > > References
> > >
> > > [1] Attend-and-excite: Attention-based semantic guidance for text-to-image diffusion models
> > >
> > > [2] Training-Free Consistent Text-to-Image Generation

---

### Official Review · Reviewer_gJqQ · 2025-07-03

**Clarity:** 3
**Significance:** 3
**Originality:** 3
**Rating:** 4
**Confidence:** 3

**Summary:**

This paper introduces a training-free inference-time guidance method to improve temporal coherence in text-to-video diffusion models. The key insight is that temporally coherent motion should exhibit low patch-wise variance across frames in the model's latent space.
The paper computes appearance-debiased temporal signals by measuring L1 distances between consecutive latent frames. Then the paper measures the motion coherence by calculating patch-wise variance across the temporal dimension as a proxy. By gradient-based optimization of the input latent during specific denoising timesteps, the maximal patchwise variance is minimized and thus the temporal coherence is improved.
The method is evaluated on Wan2.1-1.3B and CogVideoX-5B models, showing consistent improvements in motion coherence while maintaining visual quality and text alignment.

**Questions:**

1. Timestep Generalization: How sensitive is the method to the choice of optimization timesteps? Have you explored adaptive timestep selection based on the model's temporal dynamics?
2. Computational Efficiency: Have you investigated more efficient optimization strategies (e.g., fewer gradient steps, approximate gradients) that could reduce the computational overhead while maintaining effectiveness?

**Ethical Concerns:**

["NO or VERY MINOR ethics concerns only"]

**Final Justification:**

Thanks for the author's response. Most of my concerns are resolved and I will maintain my initial score as borderline accept.

**Limitations:**

yes

**Paper Formatting Concerns:**

No obvious formatting issues

**Quality:**

3

**Strengths And Weaknesses:**

Strengths

1. The insight of the paper is valuable. The core insight that temporal coherence correlates with low patch-wise variance in latent space is intuitive and well-supported by both qualitative and quantitative analysis. This finding is well-supported by experiments in the Section3.2.
2. The proposed method is training-free, making it a practical plug-and-play solution for existing models.
3. The evaluation is comprehensive. The paper includes thorough experiments with both automatic metrics (VBench) and human evaluations, showing consistent improvements across multiple criteria.


Weaknesses
1. The method would cause a significant 2.39× increase in inference time due to gradient computation and optimization steps.
2. The choice of maximizing over patch-wise variances (Equation 8) appears somewhat ad-hoc. While ablations show it works better than mean aggregation, the theoretical justification could make the paper stronger.
3. The selection of timesteps 1-12 for optimization is based on empirical observation rather than principled analysis. Different models or architectures might require different timestep ranges.

---

> ### Author Rebuttal · Authors · 2025-07-30
>
> We thank the reviewer for the thoughtful feedback and for recognizing our core insights, design, and evaluation.
>
> ## Runtime speedup
>
> In response to the reviewer’s feedback, we made small implementation changes that preserve correctness. Specifically, we replaced redundant operations and removed unnecessary computation (e.g., replacing loops over tensors with built-in PyTorch functions). The code changes did not change the results – we verified on all samples from our project’s website that the obtained outputs are bit-exact identical before and after the change. These optimizations led to a x1.29 speedup, reducing the mean runtime from 234.30s to 180.67s. As a result, FlowMo now incurs an overhead of only x1.82 (down from x2.13) relative to the baseline.
>
> ## The choice of maximal patch wise variance
>
> We provide both intuitive and theoretical justifications for the choice of the max operator in our optimization.
>
> Intuitively, artifacts are typically a local phenomenon that is dominated by specific spatiotemporal locations in which the patch statistics are abnormal (compared to what exists in the training set). Using the mean operator dilutes the impact of these anomalies, since it is dominated by the majority of the patches. However, their presence can be captured by a max operator, making it a natural choice to detect and mitigate inconsistencies.
>
> Theoretically, the following lemmas justify optimizing the maximum patch-wise variance to keep generated latents in-distribution.
>
> **Lemma 1**. *Noising of real videos keeps patch-wise temporal variance bounded by $ 1 + V_{\max} $, where $ V_{\max} $ is a dataset-dependent constant that bounds patch-wise variance for all patches of all videos in the training data.*
>
> First, note that $ V_{\max} $ exists since the model was trained on a finite dataset of real-valued latents. Furthermore, assuming the dataset consists of natural videos, and since VAEs minimize latent variance, $ V_{\max} $ is likely to be relatively small.
>
> Now, recall that noising a latent of a real video $ z_0 $ to timestep $ t $ is done by:
>
> $$ z_t = \sqrt{\alpha_t} z_0 + \sqrt{1 - \alpha_t} \epsilon \quad \text{where } \epsilon \sim \mathcal{N}(0, I) $$
>
> Thus, the temporal variance of each patch $ ij $ is given by:
>
> $$ V(z_t[ij]) = \alpha_t V(z_0[ij]) + (1 - \alpha_t) = 1 + \alpha_t (V(z_0[ij]) - 1) \leq 1 + V_{\max} $$
>
> Hence the maximum patch-wise variance of all samples during training is bounded.
>
> **Lemma 2**. *Consider the denoising process starting from pure Gaussian noise. Suppose that in early timesteps (i.e., when $ \alpha_t $ is small), each patch of the denoised latent remains approximately Gaussian but with unknown variance due to guidance, decoder mismatch, or score error, i.e., $ z_{k, i, j} \sim \mathcal{N}(0, s_{ij}) $, with $ s_{ij} $ varying across space. **Then**, the maximal patch-wise variance is $ \Omega\left( \frac{\log(W \cdot H)}{F} \cdot \max s_{ij} \right) $ w.h.p., where $F,W,H$ are the number of frames, width, and height of the latent, respectively.*
>
> We note that the assumptions in Lemma 2 are empirically satisfied in early denoising timesteps: the denoised latent remains approximately Gaussian, and patch-wise variances exhibit significant spatial variability consistent with the stated form.
>
> Intuitively, the lemma holds because at high noise levels, the per-pixel variance distribution is heavy-tailed, so outliers are likely to occur. Formally, given the assumptions in the lemma, the temporal variance of each patch is distributed $ V(z_t[ij]) \sim \frac{s_{ij}}{F - 1} \cdot \chi_{F - 1}^2 $, where $ F $ is the number of latent frames. The maximal patch-wise variance is thus the maximum over such $ \chi^2 $ distributions, which lies in the Gumble domain and is known to satisfy the property in the lemma.
>
> Hence, as the number of patches $ W \cdot H $ grows, local outliers (i.e., patches with high patch-wise variance) appear with non-negligible probability.
>
> **Lemma 3**. *A global stability condition for the numerical integration step in the denoising process requires that $ \eta_t \Vert J_{\max} \Vert \sqrt{(F-1) s_{\max}} \leq C $, where $J_{ij}=\frac{\partial \epsilon_\theta}{\partial Z_{ij}}$ is the Jacobian at patch $ij$, $s_{ij} $ is its patch-wise variance, and $C$ is the Courant number.*
>
> Let $ z_{t-h} \approx z_t - \eta_t JZ $ be a linearization of the denoising step.
>
> A stability condition for solving the ODE numerically is the CFL condition, which requires every patch $ij$ to satisfy $\eta_t \Vert J_{ij} \Vert \leq C $. Otherwise, the numerical process could be expensive and not converge. If the CFL condition holds, the local step magnitude satisfies:
>
> $$ \Vert z_{t-h}[ij] - z_t[ij] \Vert \leq \eta_t \Vert J_{ij} \Vert \Vert z_{ij} - \bar{z}_{ij} \Vert $$
>
> $$ = \eta_t  \Vert J_{ij}  \Vert \sqrt{(f-1) s_{ij}} $$
>
> Therefore, a global stability condition is as mentioned in the lemma. If a single patch attains a large $ s_{ij} $, the product above can cross the stability boundary, causing a local overshoot that pushes that region off the training manifold.
>
> **Corollary**. In training, patch-wise variance is bounded. However, when denoising pure Gaussian noise, high patch-wise variance is a likely local phenomenon, which may drive generation out-of-distribution. Minimizing the maximum patch-wise variance is thus a practical strategy to improve ODE stability and ensure generated latents $ z_t $ remain in-distribution.
>
> ## Selection of timestamps 1-12
>
> We justify the selection of timesteps 1-12 on both empirical and theoretical grounds.
>
> We break down the question into two: why we started optimizing from the first timestep, and why we stopped after 12. First, notice that Lemma 2 above shows that large patch-wise variance, a precursor to temporal incoherence, can arise as soon as the latent remains close to Gaussian—i.e., from the very first timestep—so optimization should start as early as the first step.
>
> As to why we stopped after 12 steps, section 3.2 shows, qualitatively and quantitatively, that coarse structure and global motion are set within the first dozen denoising steps. Figure 3 further indicates that these high-noise, “coarse” steps largely determine overall motion, making them optimal to apply FlowMo. This mirrors findings in image-based works (e.g., [1], [3]) which likewise identify early steps as critical for establishing coarse features. Furthermore, the same 1-12 range improves motion coherence on two very different models—Wan2.1 and CogVideoX (Secs. 4.1 – 4.2)—demonstrating robustness across architectures and datasets.
>
> ## Adaptive timestep selection
>
> Thank you for this valuable suggestion. First, we note that the use of a fixed intervention range is the standard practice in all other inference-time techniques we are aware of (e.g., [1], [2]). Nonetheless, adaptive selection is a promising direction for improving both runtime and output quality. In response to the review, we implemented a simple adaptive intervention strategy: if the maximum variance at a given step fell below 60 times the mean variance, we stopped the intervention early. The threshold of 60 was chosen based on the typical ratio observed at step 12 after a full 12-step intervention. This adaptive approach led to early stopping in most cases—typically between steps 8 and 10—and reduced average runtime by 20%. While performance metrics were slightly lower than in the fixed 12-step setup, we believe this initial result shows promise, and expect that more refined adaptive strategies could yield better results with relatively little effort.
>
> | Dimension | Baseline | + FlowMo | Adaptive timesteps |
> |---|---|---|---|
> | Motion Smoothness| 96.43% | **98.56%** | 97.25% |
> | Dynamic Degree | **83.21%**  | 81.96% | 82.04%  |
> | Semantic Score | 84.70% | **89.11%** | 86.12% |
> | Quality Score | 65.58% | **73.58%** | 70.11% |
> | Final Score | 75.14% | **81.34%** | 78.11% |
>
> Additionally, initial experiments on a subset of the data suggest that interventions applied between 8 and 15 frames result in metric values within 1% of those from the fixed setup—substantially better than the 30-frame setting (see ablations in our response to reviewer MYjg or the no-intervention baseline).
>
> ## Efficient optimization strategies (e.g., fewer gradient steps, approximate gradients)
>
> Following the review, we implemented approximate gradients (finite differences and random directional derivatives), but found that these lead to a substantial drop in video quality. Developing more efficient optimization techniques without compromising quality could be an interesting direction for future work.
>
> Regarding the number of gradient steps: please note that FlowMo uses a single gradient step (L192–200).
>
> Regarding implementation efficiency, please see above.
>
> ---
>
> We would be happy to engage further and clarify any additional questions.
>
>
> **References**
>
> [1] Attend-and-Excite: Attention-Based Semantic Guidance for Text-to-Image Diffusion Models
>
> [2] FreeInit: Bridging Initialization Gap in Video Diffusion Models
>
> [3] Image Generation from Contextually-Contradictory Prompts

---

### Comment · Area_Chair_zABS · 2025-08-04
**Please Read the Rebuttal and Discuss**

Dear Reviewers gJqQ, mc6Y. tGek

The authors have submitted their rebuttal.

Please carefully review all other reviews and the authors’ responses, and engage in an open exchange with the authors.

Kindly post your initial response as early as possible within the discussion window to allow sufficient time for interaction.

Your AC

---

### Note · Authors · 2025-08-12

Following discussions with the reviewers, we believe we have addressed the main concerns and provided strong evidence for FlowMo’s novel, training-free contribution to motion coherence in video generation.

All reviewers acknowledged our core insight that temporally coherent motion correlates with low patch-wise variance in latent space; Reviewer gJqQ called it “valuable and well-supported,” MYjg praised our “interesting and convincing” analyses, tGek noted our systematic validation (Sec. 3.2), and mc6Y highlighted its “generalizability.”

We addressed the primary concern about runtime overhead via implementation improvements, achieving a 1.29× speedup and cutting overhead from 2.39× to 1.82×, now competitive with other inference-time methods, e.g. Attend-and-Excite (1.73×) and ConsiStory (2×). Outputs remain bit-exact identical, confirming efficiency gains without quality loss. Adaptive timestep selection further reduces overhead to 1.58× with minimal performance impact.

We refuted concerns that temporal consistency gains reduce dynamic motion. FlowMo improves Quality Score by 8–11% while lowering Dynamic Degree by <1.5%, negligible compared to FreeInit’s 16.2% drop.

A theoretical framework is presented: three lemmas showing that optimizing max patch-wise variance keeps latents in-distribution, prevents instability, and removes local artifacts.

Ablations across all design choices (mean vs. max, debiasing, timestep ranges) show FlowMo improves Semantic Score by 3.99%, Quality Score by 6.07%, and Final Score by 5.03% over the best ablation. Cross-model validation on architecturally distinct models (Wan2.1 and CogVideoX) confirms robustness, with CogVideoX showing an 11.28% Quality Score gain.

On baseline performance gaps, we documented technical constraints: Wan2.1-1.3B’s 720p is unsupported in the public release (per the official repository’s note). Even at 480p, FlowMo surpasses official 720p Motion Smoothness metrics (98.56% vs. 97.44%), demonstrating that our method’s effectiveness transcends resolution limitations.

All reviewers positively engaged with our rebuttals: tGek said concerns were addressed and updated their score; mc6Y noted issues were “well-addressed” and considered a higher rating; MYjg confirmed “most concerns are addressed.” This broad recognition of FlowMo’s training-free, plug-and-play practicality, backed by theory and experiments, solidifies it as key in tackling video generation’s core challenge: temporal coherence.

---

### Decision · Program_Chairs · 2025-09-17

**Decision:**

Accept (poster)

**Comment:**

This paper introduces a practical, training-free method for improving temporal coherence in video diffusion models. The core idea of minimizing patch-wise variance is well-motivated, theoretically grounded, and validated across multiple datasets and models. The reviewers appreciated the clarity, systematic analysis, and cross-model generalization.

The main weakness is runtime overhead. While the authors reduced it from 2.39× to 1.82× (and to 1.58× with adaptive stopping), some reviewers (notably MYjg) still find this overhead problematic for real-world applications. Reviewer mc6Y expressed concerns about the potential reduction of dynamic degree. He/she also noted a baseline discrepancy with official Wan2.1 performance. The authors attributed this to resolution limitations in the released model. Concerns about ablations and theoretical justification were largely resolved during rebuttal.

Reviewer tGek recommended acceptance after rebuttal. Reviewer mc6Y recommended borderline accept as the concerns about dynamic degree is resolved, but emphasized that the authors should compare with higher-performing implementation of the WAN baseline. gJqQ and MYjg both maintained borderline accept due to runtime overhead, though they acknowledged that the authors addressed most other concerns.

Overall, the paper is seen as a technically solid, practical contribution, albeit with some remaining efficiency and evaluation concerns. The runtime concern, while valid, the AC believes that it is sufficiently mitigated, and the work will be of interest to the video generation community.